# 🔆 VIDEOMIND: A CHAIN-OF-LORA AGENT FOR TEMPORAL-GROUNDED VIDEO REASONING

**Ye Liu**[1][*], **Kevin Qinghong Lin**[2][*], **Chang Wen Chen**[1][✉], **Mike Zheng Shou**[2][✉]
[1]The Hong Kong Polytechnic University [2]National University of Singapore
`coco.ye.liu@connect.polyu.hk`

## ABSTRACT

Videos, with their unique temporal dimension, demand precise grounded understanding, where answers are directly linked to visual, interpretable evidence. Despite significant breakthroughs in text-based reasoning with large language models, multi-modal reasoning – especially for videos – remains limited. In this work, we fill this gap by introducing **VideoMind**, a novel video-language agent for temporal-grounded video reasoning. Our method involves two key innovations: (1) We identify four essential capabilities for grounded video reasoning and propose a role-based agentic workflow, comprising a `planner` to coordinate roles, a `grounder` for temporal event localization, a `verifier` to assess event candidates, and an `answerer` for question answering. (2) To efficiently integrate these roles during inference, we propose a novel **Chain-of-LoRA** mechanism, where a unified base model with multiple LoRA adapters is leveraged to enable seamless role switching, balancing efficiency and flexibility. Extensive experiments on 15 benchmarks across Grounded VideoQA, Video Temporal Grounding, and General VideoQA tasks demonstrate the effectiveness of the proposed scheme in advancing video agent, test-time scaling, and long-form video reasoning. Code, models, datasets, and demos are available at https://videomind.github.io/.

## 1 INTRODUCTION

Recent advancements in large language models (LLMs) have demonstrated remarkable success in text-based reasoning (Wei et al., 2022; Yao et al., 2023a; Shinn et al., 2023), significantly improving both accuracy and interpretability in complex problem-solving scenarios (Yao et al., 2023b). Following these breakthroughs, efforts have been devoted to extending these reasoning capabilities to multi-modal domains (Seed, 2026; Xu et al., 2025; Thawakar et al., 2025) such as vision-centric science (Lu et al., 2022) and math (Ma et al., 2025) understanding.

Among multi-modal signals, videos pose a unique challenge due to their temporal dimension, introducing complexities absent in images or text. Effective video reasoning requires not only recognizing visual appearances but also understanding how they evolve over time (Xiao et al., 2024; Chen et al., 2024a; Liu et al., 2024e; Wu et al., 2025). While recent visual Chain-of-Thought (CoT) methods (Zhang et al., 2023c; Xu et al., 2025; Thawakar et al., 2025) excel at generating detailed thoughts for static images, they struggle with long videos as they cannot explicitly localize or revisit earlier parts of the sequence, as presented in Figure 1 (left). Humans, by contrast, can reason over long videos with ease (Xie et al., 2025b; 2024b; 2022): they break down complex problems, identify relevant moments, revisit them to confirm details, and synthesize their observations into coherent answers. This natural proficiency motivates the development of an AI agent that emulates this process – flexibly coordinating multiple capabilities to achieve advanced, vision-centric reasoning.

In this work, we introduce **VideoMind**, a video-language agent with enhanced temporal-grounded reasoning capabilities. To meet the demands of diverse tasks, we define four essential roles for understanding complex long-form videos: (1) a **planner** to decompose tasks and coordinate other roles, (2) a **grounder** for precise moment localization, (3) a **verifier** for moment candidates assessment, and (4) an **answerer** for moment-aware response generation. Each role is carefully

---

[*]Equal contribution.    [✉]Corresponding authors.

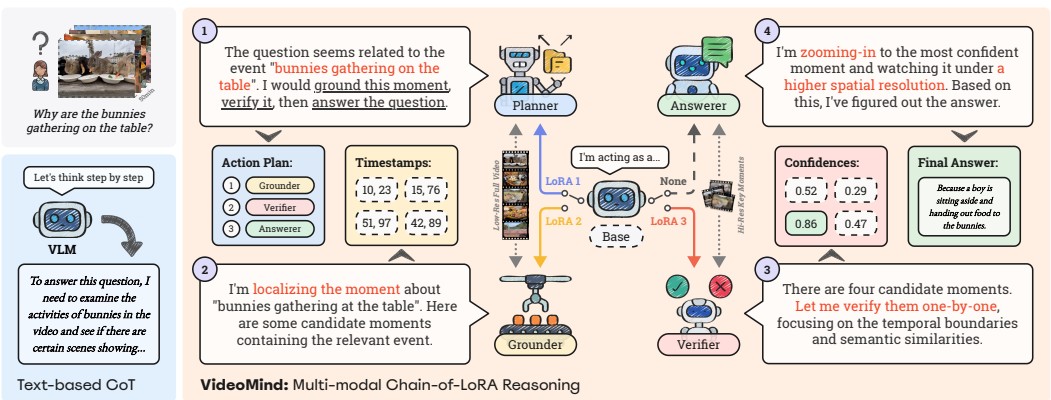

Figure 1: Illustration of VideoMind's Chain-of-LoRA reasoning mechanism. The problem is decomposed by the planner and distributed to the grounder, verifier, and answerer to systematically localize, verify, and interpret the relevant video moments.

designed to deliver strong performance, for example, the grounder is equipped with a timestamp decoder to ensure accurate temporal grounding. To enable efficient integration of these roles, we also propose a novel **Chain-of-LoRA** mechanism, where all the roles are implemented based on a unified LMM backbone with role-specific LoRA adapters (Hu et al., 2022). Therefore, role-specific capabilities can be trained separately on tailored datasets. During inference, all the LoRA parameters are cached into the memory, so that each role could be activated by simply switching to the corresponding LoRA, as shown in Figure 1 (right). This approach reflects a minimalist yet flexible design philosophy, facilitating seamless transitions and interactions among roles without incurring the memory overhead of maintaining multiple full models. As a result, VideoMind achieves both efficiency and flexibility on diverse video understanding tasks.

We conduct extensive experiments on 15 public benchmarks, including 4 on Grounded VideoQA, 6 on Video Temporal Grounding, and 5 on General VideoQA, to evaluate the effectiveness of our approach. VideoMind exhibits strong adaptability in addressing diverse reasoning tasks by jointly providing accurate responses and temporal-grounded evidence. Notably, our 2B model surpasses GPT-4o (OpenAI, 2024a) and Gemini-1.5-Pro (Reid et al., 2024) on several long video benchmarks such as CG-Bench (Chen et al., 2024a), MLVU (Zhou et al., 2024), and LVBench (Wang et al., 2024c). State-of-the-art performance is also achieved on temporal grounding datasets including QVHighlights (Lei et al., 2021) and Charades-STA (Gao et al., 2017). We further conduct ablation studies to justify our design choices, particularly the Chain-of-LoRA mechanism for enhancing flexibility while preserving efficiency. Our contributions are summarized as follows:

1. We propose **VideoMind**, a multi-modal agentic framework that enhances video reasoning by emulating human cognitive processes, including task decomposition, moment localization and verification, and answer synthesis. It addresses the unique challenges of long video reasoning in a progressive and structured manner.

2. We introduce **Chain-of-LoRA**, an efficient test-time scaling mechanism that enables a single model to seamlessly switch among multiple roles. This approach enhances Video-Mind's flexibility without incurring additional memory overhead.

3. Our method demonstrates strong performance across three scenarios: Grounded VideoQA, Video Temporal Grounding, and General VideoQA. Notably, VideoMind-2B outperforms GPT-4o and Gemini-1.5-Pro on several long video benchmarks.

## 2 RELATED WORK

**Temporal-grounded Video Understanding**  Significant advances in video understanding have propelled tasks such as video retrieval (Lin et al., 2022; Lin & Shou, 2025) and captioning (Zhao et al., 2023; Lin et al., 2024b; Chen et al., 2024b). However, these models often lack *visually grounded correspondence* and interpretability, particularly for long-form videos. The task of Video Temporal Grounding (Gao et al., 2017; Krishna et al., 2017) tackles this issue by requiring precise

temporal localization for diverse queries, though regression-based models (Liu et al., 2022; 2024d) excel at localization but fall short in providing textual interpretability. Recent benchmarks (Xiao et al., 2024; Chen et al., 2024a; Liu et al., 2024e) intensify this challenge, demanding both reasoning for complex questions and fine-grained temporal correspondence. Previous baselines for these tasks typically rely on multi-task objectives or modular agents composed of distinct components (Wang et al., 2024d; Fan et al., 2024), often yielding sub-optimal performance or overly complex systems, which constrain their efficiency and flexibility. Our VideoMind is an agentic workflow built upon a unified LMM, seamlessly integrating multiple functionalities while enhancing localization and interpretability, thus surpassing the limitations of prior methods.

**Multi-modal Reasoning** Large Multi-modal Models (Liu et al., 2023; 2025) exhibit generalized capabilities such as free-form question answering. However, they fall short in addressing complex challenges that often require reasoning (Wei et al., 2022). One approach to overcome this is to develop agent-based interfaces (Zhang et al., 2023a; Kahatapitiya et al., 2024), which integrates textual outputs from visual tools to enable reasoning via LLMs. Advanced methods (Suris et al., 2023; Yang et al., 2023; Gao et al., 2023) invoke visual APIs through progressive execution and reasoning. Alternatively, pure text-based reasoning (OpenAI, 2024b; Guo et al., 2025) has been a dominant paradigm in LLMs, exemplified by training with long CoT processes using reinforcement learning, which provides detailed step-by-step reasoning, with some works (Chen et al., 2025b; Feng et al., 2025) extending this mechanism to the visual domain. Despite these advances, extending reasoning to videos remains an open challenge. Given the long-context nature of informative videos, we believe that *a vision-centric* CoT should incorporate a human-like (Xie et al., 2023; 2024a; 2025a; Wen et al., 2025) re-watching strategy and self-validation of intermediate observations, leading us to introduce a novel Chain-of-LoRA framework for video reasoning.

**Inference-time Searching** Inference-time searching has emerged as a critical technique for tackling complex reasoning challenges in different domains. The advent of OpenAI o1 (OpenAI, 2024b) has advanced these inference-time techniques within LLMs by integrating sampling strategies such as controlled decoding (Chakraborty et al., 2024), Best-of-N sampling (Lightman et al., 2023), and Monte Carlo Tree Search (MCTS) (Wang et al., 2024e), allowing LLMs to iteratively refine outputs and achieve superior performance without altering their underlying weights. However, the potential of inference-time searching remains largely untapped in video understanding, where temporal reasoning introduces unique challenges. In our framework, we explore how such a strategy can be tailored for video temporal reasoning, observing that models are highly sensitive to the selection of temporal segments, often producing unreliable predictions when segment choices are sub-optimal. To address this, we propose a *moment-level* searching approach where a grounder generates multiple candidates, followed by a verifier that evaluates and determines the correct correspondence. The framework also supports flexible role switching with minimal memory overhead.

## 3 METHOD

**Overview** Figure 2 provides an overview of VideoMind. Our model derives from the Qwen2-VL (Wang et al., 2024b) architecture, consisting of an LLM backbone and a ViT-based visual encoder support dynamic resolution inputs. Given a video input $\mathcal{V}$ and a text query $\mathcal{Q}$, the model performs step-by-step reasoning by adaptively calling different roles: (1) **Planner**: Dynamically coordinates the following roles based on the query. (2) **Grounder**: Identifies and localizes relevant video moments. (3) **Verifier**: Evaluates the validity of the moments identified by the grounder, refining them through a zoom-in process with boolean outputs. (4) **Answerer**: Generates the final response in natural language. This mechanism enables the models to **revisit the videos several times** (with varying temporal segments & spatial resolutions) to derive the final response.

### 3.1 PLANNER

An agent must be flexible enough to handle diverse tasks and efficiently determine which functions (roles) to call. To achieve this, we design the **planner**, which dynamically coordinates all the other roles for each query. It decides the sequence of function calls based on the multi-modal context. We utilize a JSON-style object `{"type": "<role>", "value": "<argument>"}` to denote a function call. In this way, a sequence of roles can be succinctly represented as a list of such objects. Three reasoning plans for different tasks are pre-defined and illustrated in Figure 3.

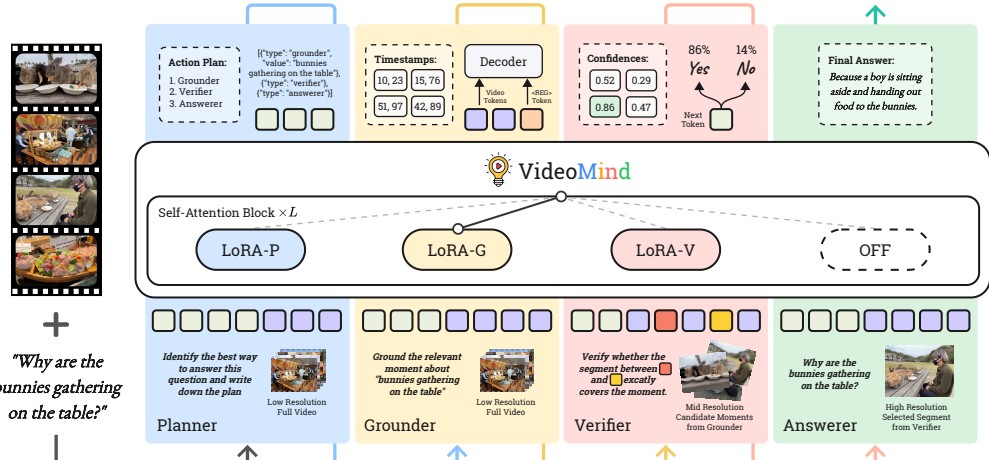

Figure 2: The overall workflow of VideoMind. Given a video and a query, it adaptively activates different roles (*e.g.*, `Planner` → `Grounder` → `Verifier` → `Answerer` in this case) and performs step-by-step reasoning by calling individual modules.

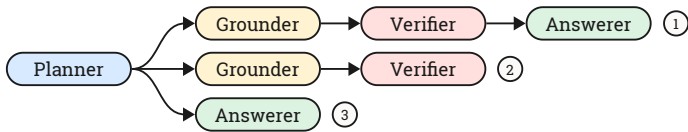

Figure 3: Planner coordinates all the other roles based on the video and query context, offering three reasoning plans and a query rephrasing mechanism to address diverse demands.

**(1) Grounding & Verifying & Answering:** This plan requires the agent to generate both a textual response and a corresponding temporal moment. For example, in Grounded VideoQA scenarios (Xiao et al., 2021), to answer the question *"What is the boy doing when the baby is crying?"*, the agent should identify the moment of *"baby is crying"*, and then investigate the boy's activity.

**(2) Grounding & Verifying:** This plan is designed for grounding-only tasks such as moment retrieval (Lei et al., 2021; Gao et al., 2017). For questions like *"When does the woman go downstairs?"*), the model should provide precise timestamps directly as the answer. Since the grounding results could potentially be unreliable, an extra zoom-in verification step is necessary.

**(3) Answering Only:** If the question is straightforward (*e.g.*, *"Summarize this video"*) or the video is very short (*e.g.*, less than 10s), it could be unnecessary to perform grounding. Instead, the model should watch the entire video and answer the question directly.

**Query Rephrasing**   When the user query lacks sufficient detail for accurate moment localization, the planner is allowed to **rephrase** the question into a more descriptive version. For instance, the question *"What is the person sitting on the bed doing as the baby plays?"* may confuse the grounder as it contains multiple events (*"person sitting on the bed"* and *"baby plays"*). It can be rephrased to *"the baby is playing"* as an accurate scene description.

To train the planning and query rephrasing capabilities, we curated a dataset of 39K samples (shown in Table 1) from public benchmarks. For planning, we aligned each reasoning plan with corresponding question types: *temporal* questions from NExT-QA (Xiao et al., 2021) are assigned to Plan-1, moment queries from QVHighlights (Lei et al., 2021) are for Plan-2, and *causal* & *descriptive* questions from NExT-QA (Xiao et al., 2021) are for Plan-3. For query rephrasing, we leverage GPT-4o mini (OpenAI, 2024a) to generate synthetic `video` + `question` → `query` samples for training.

## 3.2   GROUNDER

The **grounder** aims to localize relevant moments (*i.e.*, predicting start and end timestamps) based on text queries, thereby supporting the reasoning process by identifying visual cues. This requirement calls for the development of an LMM with robust temporal grounding capabilities.

**Timestamp Decoder**  Instead of directly predicting timestamps through language modeling (Ren et al., 2024) or special tokens (Huang et al., 2024a; Liu et al., 2024e), we develop a timestamp decoder to maximize the LMM-based grounding performance. Specifically, we introduce a `<REG>` token to facilitate this process. When the `<REG>` token is generated, the last-layer hidden states of it and all the visual tokens will be sent into the decoder for timestamp prediction, obtaining a tuple $[t_{start}, t_{end}]$ representing the normalized start and end timestamps.

As shown in Figure 4, the decoder accepts the hidden states of the visual tokens $\mathbf{h}_v \in \mathbb{R}^{(T \times H \times W) \times D_L}$ and the `<REG>` token $\mathbf{h}_r \in \mathbb{R}^{1 \times D_L}$ as inputs, where $T$, $H$, $W$, $D_L$ are the downsampled number of frames, height, width, and hidden dimensions of the LLM, respectively. We apply a 1D average pooling with kernel size and stride equal to $H \times W$ to compress the visual tokens to one token per frame.

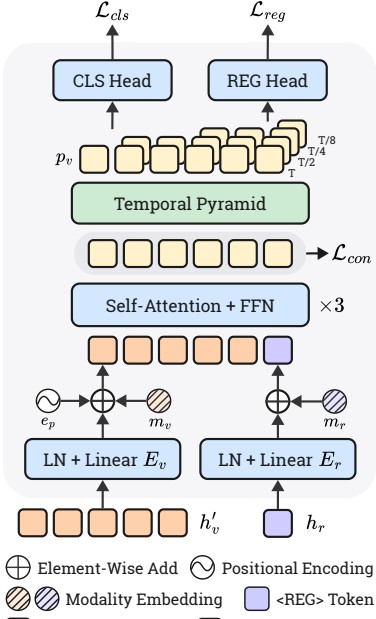

$$\mathbf{h}'_v = \text{AvgPool}(\mathbf{h}_v) \in \mathbb{R}^{T \times D_L} \quad (1)$$

Then, $\mathbf{h}'_v$ and $\mathbf{h}_r$ are projected by two linear layers $E_v$ and $E_r$ to reduce the hidden dimension to $D$.

$$\mathbf{e}_v = E_v(\mathbf{h}'_v) \in \mathbb{R}^{T \times D}, \quad \mathbf{e}_r = E_r(\mathbf{h}_r) \in \mathbb{R}^{1 \times D} \quad (2)$$

The resulting $\mathbf{e}_v$ and $\mathbf{e}_r$ serve as consolidated representations of the video frames and the query[1], respectively. To effectively integrate their information, we concatenate them along the sequence dimension and send them into a three-layer transformer encoder (Vaswani et al., 2017).

$$[\mathbf{e}'_v; \mathbf{e}'_r] = \text{Transformer}([\mathbf{e}_v + \mathbf{m}_v + \mathbf{e}_p; \mathbf{h}_r + \mathbf{m}_r]) \quad (3)$$

Here, modality indicators $m_v \in \mathbb{R}^{1 \times D}$ and $m_r \in \mathbb{R}^{1 \times D}$ are randomly initialized learnable embeddings. $m_v$ is expanded to $T \times D$ before being added with $e_v$. $e_p$ is a normalized sinusoidal positional encoding (Vaswani et al., 2017) for preserving temporal awareness. The output sequence is split back into $\mathbf{e}'_v$ and $\mathbf{e}'_r$, indicating the contextualized frame and query embeddings, respectively.

Figure 4: Detailed architecture of the timestamp decoder.

**Temporal Feature Pyramid**  To improve the model's adaptability to videos and moments of varying lengths, we map $\mathbf{e}'_v$ into a four-level temporal feature pyramid (Liu et al., 2024d; Zhang et al., 2022). Each level is produced by a `Conv1D → LayerNorm → SiLU` block, where the `Conv1D` employs a kernel size and stride of 2. Therefore, the resulting four levels retain 1, 1/2, 1/4, and 1/8 of the original sequence length, respectively. To accelerate the prediction, we concatenate the sequences from all pyramid levels along the temporal dimension to form $\mathbf{p}_v$ with length $L = T + T/2 + T/4 + T/8$, allowing parallelized prediction across temporal resolutions.

**Prediction Heads**  We introduce two heads for timestamps prediction: **(1) A classification head** is designed for frame-level foreground-background classification. This is instantiated by a two-layer `Conv1D` module with kernel size 3 and padding 1, followed by a Sigmoid activation. The outputs are frame-level confidence scores $\{\hat{c}_i\}_{i=0}^{L}$ indicating whether each frame falls inside the desired moment. A binary focal loss (Lin et al., 2017) is utilized to optimize these scores.

$$\mathcal{L}_{cls} = -\lambda_{cls}\alpha(1 - \hat{c}_i)^{\gamma} \log(\hat{c}_i) \quad (4)$$

Here, $\alpha = 0.9$ and $\gamma = 2.0$ are hyperparameters of the focal loss, and $\lambda_{cls}$ is the loss reweighing term. **(2) A boundary regression head** is adopted to predict the frame-level temporal offsets for start and end boundaries $\{[\hat{b}_i^s, \hat{b}_i^e]\}_{i=0}^{L}$. This is also a two-layer `Conv1D` block (with 2 output channels), followed by an exponential activation. Predictions from different pyramid levels are further modulated by different learnable scaling factors. These outputs are supervised by an $L1$ loss.

$$\mathcal{L}_{reg} = \lambda_{reg}(|b_i^s - \hat{b}_i^s| + |b_i^e - \hat{b}_i^e|) \quad (5)$$

In order to realize better alignment between $e'_v$ and $e'_r$, we incorporate an additional contrastive loss to encourage learning more discriminative representations. Specifically, we calculate the cosine

---

[1]We use the term "query" to denote the features of `<REG>` token.

Table 1: Training datasets for different roles. Source datasets were repurposed for training planner and verifier. *mr* and *step* denote the moment retrieval and step localization subsets, respectively.

| Role | #Samples | Source Datasets |
|---|---|---|
| Planner | 39K | NeXT-QA (34K), QVHighlights (5K) |
| Grounder | 210K | QVHighlights (5K), DiDeMo (33K), TACoS (9K), InternVid-VTime (54K), CosMo-Cap (87K), QuerYD (19K), HiREST$_{mr}$ (8K), HiREST$_{step}$ (4K) |
| Verifier | 232K | DiDeMo (165K), TACoS (43K), QVHighlights (24K) |

similarities among all frame-query pairs (denoted as $\{s_i\}_{i=0}^{L}$), then sample a positive frame (falling within the ground truth boundary) and apply the following optimization objective:

$$\mathcal{L}_{con} = -\lambda_{con} \log \frac{\exp(s_p/\tau)}{\exp(s_p/\tau) + \sum_{i \in \Theta} \exp(s_i/\tau)} \tag{6}$$

Here, $\Theta$ is the set of frame indices with $s_p > s_i$, and $\tau = 0.07$ is the temperature parameter. The final loss for the timestamp decoder is the sum of these losses at all layers with $\lambda_{cls} = 5.0$, $\lambda_{reg} = 1.0$, and $\lambda_{con} = 0.05$. The training datasets for the grounder are listed in Table 1.

### 3.3 VERIFIER

Key moments are crucial for providing visual cues, yet they might be unreliable due to grounding errors. Thus, further verifications are necessary. We let the grounder generate top-5 predictions, then employ the **verifier** to select the most reliable one. This process is presented below.

**Recap by Zooming-in** For each candidate moment, we apply a zoom-in strategy by expanding the boundaries by 50% on both sides and temporally cropping the enlarged segment. The resulting segment and the original text query are sent to the verifier to assess whether the queried event exactly occurs within the temporal boundaries. To enhance boundary awareness, we adopt two special tokens, <SEG-START> and <SEG-END>, to explicitly mark the beginning and end of the moment. These tokens are inserted among the visual tokens at the corresponding frames, effectively guiding the model in recognizing moment boundaries.

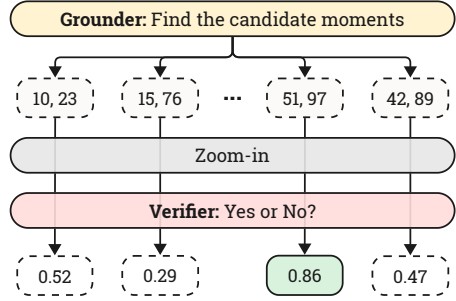

Figure 5: The grounder generates multiple candidate moments, which are then refined by the verifier via **zooming-in** to investigate and select the best one.

**Boolean Judgement** The verifier's responses are binary, *i.e.*, either *"Yes"* or *"No"*. To train this role, we sample predictions from the grounder and assign binary labels based on an IoU threshold of 0.5. The model is then fine-tuned via SFT to predict these labels. During inference, for each candidate moment, we employ teacher forcing to obtain the likelihoods of the <Yes> and <No> tokens, denoted as $L_y$ and $L_n$, respectively. The confidence score is then computed as $\text{Sigmoid}(L_y - L_n)$. The moment with the highest score is selected and passed to the answerer.

### 3.4 ANSWERER

The **answerer** responds to the given question based on the cropped video segment (w/ grounder) or the whole video (w/o grounder). Since the objective of this role is strictly aligned with existing LMMs, we employ the original model directly *without fine-tuning or architectural modifications*.

### 3.5 CHAIN-OF-LORA

The four roles introduced above demonstrate distinct yet complementary capabilities, collaborating to achieve advanced vision-centric reasoning. However, simply integrating these roles into a single model poses challenges, as their core functionalities can interfere with one another. To avoid inefficiently implementing them as multiple models while still accommodating diverse demands, we propose a novel **Chain-of-LoRA** mechanism to enable flexible and efficient role switching.

Table 2: Performance comparison on Grounded VideoQA on CG-Bench (Chen et al., 2024a).

| Method | Size | long-acc. | mIoU | rec.@IoU | acc.@IoU |
|---|---|---|---|---|---|
| GPT-4o (OpenAI, 2024a) | – | **45.2** | 5.62 | 8.30 | 4.38 |
| Gemini-1.5-Pro (Reid et al., 2024) | – | 37.2 | 3.95 | 5.81 | 2.53 |
| Claude-3.5-Sonnet (Anthropic, 2025) | – | 40.5 | 3.99 | 5.67 | 2.79 |
| Video-LLaVA (Lin et al., 2023a) | 7B | 16.2 | 1.13 | 1.96 | 0.59 |
| VideoLLaMA (Zhang et al., 2023b) | 7B | 18.4 | 1.21 | 1.87 | 0.84 |
| VideoChat2 (Li et al., 2024b) | 7B | 19.3 | 1.28 | 1.98 | 0.94 |
| ST-LLM (Liu et al., 2024c) | 7B | 23.8 | 2.23 | 2.86 | 1.13 |
| ShareGPT4Video (Chen et al., 2024d) | 16B | 26.7 | 1.85 | 2.65 | 1.01 |
| Chat-UniVi-v1.5 (Jin et al., 2023) | 13B | 25.9 | 2.07 | 2.53 | 1.21 |
| VILA (Lin et al., 2024a) | 8B | 28.7 | 1.56 | 2.89 | 1.35 |
| LongVA (Zhang et al., 2024) | 7B | 28.7 | 2.94 | 3.86 | 1.78 |
| LLaVA-OneVision (Li et al., 2024a) | 7B | 31.1 | 1.63 | 1.78 | 1.08 |
| Video-CCAM (Fei et al., 2024) | 14B | 29.7 | 2.63 | 3.48 | 1.83 |
| Kangaroo (Liu et al., 2024b) | 8B | 30.2 | 2.56 | 2.81 | 1.94 |
| VITA (Fu et al., 2024b) | 8×7B | 33.3 | 3.06 | 3.53 | 2.06 |
| Qwen2-VL (Wang et al., 2024b) | 72B | 41.3 | 3.58 | 5.32 | 3.31 |
| InternVL2 (OpenGVLab, 2024) | 78B | 42.2 | 3.91 | 5.05 | 2.64 |
| **VideoMind** (Ours) | 2B | 31.0 | 5.94 | 8.50 | 4.02 |
| **VideoMind** (Ours) | 7B | 38.4 | **7.10** | **9.93** | **4.67** |

Table 3: Performance comparison on Grounded VideoQA on ReXTime (Chen et al., 2024c). FT indicates fine-tuning on the target dataset.

| Method | Size | FT | R@0.3 | R@0.5 | mIoU | Acc | Acc@IoU |
|---|---|---|---|---|---|---|---|
| VTimeLLM (Huang et al., 2024a) | 7B | ✗ | 28.84 | 17.41 | 20.14 | 36.16 | – |
| TimeChat (Ren et al., 2024) | 7B | ✗ | 14.42 | 7.61 | 11.65 | 40.04 | – |
| LITA (Huang et al., 2024b) | 13B | ✗ | 29.49 | 16.29 | 21.49 | 34.44 | – |
| VTimeLLM (Huang et al., 2024a) | 7B | ✓ | 43.69 | 26.13 | 29.92 | 57.58 | 17.13 |
| TimeChat (Ren et al., 2024) | 7B | ✓ | 40.13 | 21.42 | 26.29 | 49.46 | 10.92 |
| **VideoMind** (Ours) | 2B | ✗ | 34.31 | 22.69 | 24.83 | 69.06 | 17.26 |
| **VideoMind** (Ours) | 7B | ✗ | **38.22** | **25.52** | **27.61** | **74.59** | **20.20** |

In greater detail, all roles are based on a shared LMM backbone and are augmented with different LoRA adapters (Hu et al., 2022). Note that an additional timestamp decoder is used exclusively by the grounder. During inference, the framework dynamically activates role-specific LoRA adapters according to the planner, thereby maximizing the strengths of each role while minimizing the memory consumption and architectural modifications to the base model.

## 4 EXPERIMENTS

We evaluate the effectiveness of VideoMind through extensive experiments across 15 public benchmarks. Specifically, we study the following research questions.

**Q1.** Whether VideoMind is flexible and effective on diverse video understanding tasks compared to the corresponding baselines with task-specific designs?

**Q2.** Compared with (1) training a single agent on multiple tasks or (2) distributing all roles to different models, what advantages does Chain-of-LoRA offer?

**Q3.** What effects does each individual design contribute? More importantly, whether each role is necessary for building such a video reasoning system?

Detailed information about the benchmarks, evaluation settings, implementation details, and more experimental results can be found in the appendix.

### 4.1 Q1: COMPARISON WITH STATE-OF-THE-ARTS

**Grounded Video Question Answering**  Table 2 compares the Grounded VideoQA performance on CG-Bench (Chen et al., 2024a), a challenging video benchmark with an average duration of 27 minutes. On temporal grounding metrics (mIoU and rec.@IoU), our lightweight 2B model outperforms all the baselines, including GPT-4o (OpenAI, 2024a) and Gemini 1.5 Pro (Reid et al., 2024). Our 7B model further setups a new state-of-the-art on clue-grounded QA (acc.@IoU). In Table 3 and Table 4, we further present the comparison results on ReXTime (Chen et al., 2024c) and NExT-GQA

Table 4: Performance comparison on Grounded VideoQA on NExT-GQA (Xiao et al., 2024).

| Method | Size | IoU | | | IoP | | | Acc@GQA |
|---|---|---|---|---|---|---|---|---|
| | | R@0.3 | R@0.5 | mIoU | R@0.3 | R@0.5 | mIoP | |
| FrozenBiLM NG+ (Yang et al., 2022) | 890M | 13.5 | 6.1 | 9.6 | 28.5 | 23.7 | 24.2 | 17.5 |
| SeViLA (Yu et al., 2023) | 4B | 29.2 | 13.8 | 21.7 | 34.7 | 22.9 | 29.5 | 16.6 |
| LangRepo (Kahatapitiya et al., 2024) | 8×7B | – | 12.2 | 18.5 | – | 28.7 | 31.3 | 17.1 |
| VideoStreaming (Qian et al., 2024b) | 8.3B | – | 13.3 | 19.3 | – | 31.0 | 32.2 | 17.8 |
| LLoVi (Zhang et al., 2023a) | 1.8T | – | 15.3 | 20.0 | – | **36.9** | 37.3 | 24.3 |
| HawkEye (Wang et al., 2024f) | 7B | 37.0 | 19.5 | 25.7 | – | – | – | – |
| VideoChat-TPO (Yan et al., 2024) | 7B | 41.2 | 23.4 | 27.7 | 47.5 | 32.8 | 35.6 | 25.5 |
| **VideoMind** (Ours) | 2B | 45.2 | 23.2 | 28.6 | 51.3 | 32.6 | 36.4 | 25.2 |
| **VideoMind** (Ours) | 7B | **50.2** | **25.8** | **31.4** | **56.0** | 35.3 | **39.0** | **28.2** |

Table 5: Performance comparison on video temporal grounding on Charades-STA (Gao et al., 2017) and ActivityNet-Captions (Krishna et al., 2017). FT means fine-tuning on the target dataset.

| Method | Size | FT | Charades-STA | | | | ActivityNet-Captions | | | |
|---|---|---|---|---|---|---|---|---|---|---|
| | | | R@0.3 | R@0.5 | R@0.7 | mIoU | R@0.3 | R@0.5 | R@0.7 | mIoU |
| VTimeLLM (Huang et al., 2024a) | 7B | ✗ | 51.0 | 27.5 | 11.4 | 31.2 | 44.0 | 27.8 | 14.3 | 30.4 |
| TimeChat (Ren et al., 2024) | 7B | ✗ | 51.5 | 32.2 | 13.4 | – | – | – | – | – |
| Momentor (Qian et al., 2024a) | 7B | ✗ | 42.6 | 26.6 | 11.6 | 28.5 | 42.9 | 23.0 | 12.4 | 29.3 |
| ChatVTG (Qu et al., 2024) | 7B | ✗ | 52.7 | 33.0 | 15.9 | 34.9 | 40.7 | 22.5 | 9.4 | 27.2 |
| VideoChat-TPO (Yan et al., 2024) | 7B | ✗ | 58.3 | 40.2 | 18.4 | 38.1 | – | – | – | – |
| E.T. Chat (Liu et al., 2024e) | 4B | ✗ | 65.7 | 45.9 | 20.0 | 42.3 | 24.1 | 12.8 | 6.1 | 18.9 |
| Grounded-VideoLLM (Wang et al., 2024a) | 4B | ✗ | 54.2 | 36.4 | 19.7 | 36.8 | – | – | – | – |
| TRACE (Guo et al., 2024) | 7B | ✗ | – | 40.3 | 19.4 | – | – | – | – | – |
| LLaVA-ST (Li et al., 2025a) | 7B | ✗ | 63.1 | 44.8 | 23.4 | 42.4 | – | – | – | – |
| UniTime (Li et al., 2025b) | 7B | ✗ | – | **59.1** | **31.9** | **52.2** | – | 22.8 | 14.1 | 27.3 |
| **VideoMind** (Ours) | 2B | ✗ | 67.6 | 51.1 | 26.0 | 45.2 | 44.0 | 26.5 | 12.6 | 30.1 |
| **VideoMind** (Ours) | 7B | ✗ | **73.5** | **59.1** | 31.2 | 50.2 | **48.4** | **30.3** | **15.7** | **33.3** |

Table 6: Performance comparison on General VideoQA on Video-MME (Fu et al., 2024a), MLVU (Zhou et al., 2024), and LVBench (Wang et al., 2024c).

| Method | Size | Video-MME | | MLVU | LVBench |
|---|---|---|---|---|---|
| | | All | Long | M-Avg | Overall |
| GPT-4o (OpenAI, 2024a) | – | 71.9 | 65.3 | 54.5 | 30.8 |
| Gemini-1.5-Pro (Reid et al., 2024) | – | 75.0 | 67.4 | – | 33.1 |
| Video-LLaVA (Lin et al., 2023a) | 7B | 41.1 | 37.8 | 29.3 | – |
| TimeChat (Ren et al., 2024) | 7B | 34.3 | 32.1 | 30.9 | 22.3 |
| MovieChat (Song et al., 2023) | 7B | 38.2 | 33.4 | 25.8 | 22.5 |
| PLLaVA (Xu et al., 2024) | 34B | 40.0 | 34.7 | 53.6 | 26.1 |
| VideoChat-TPO (Yan et al., 2024) | 7B | 48.8 | 41.0 | 54.7 | – |
| LongVA (Zhang et al., 2024) | 7B | 52.6 | 46.2 | 56.3 | – |
| **VideoMind** (Ours) | 2B | 55.4 | 46.3 | 58.7 | 35.4 |
| **VideoMind** (Ours) | 7B | **58.2** | **49.2** | **64.4** | **40.8** |

(Xiao et al., 2024). Despite the challenges posed by the causal event relationships on ReXTime, our model can successfully identify the correct moment, resulting in significant performance boosts compared with zero-shot baselines. On NExT-GQA, compared to agent-based solutions such as LLoVi (Zhang et al., 2023a) and LangRepo (Kahatapitiya et al., 2024) and end-to-end methods like VideoChat-TPO (Yan et al., 2024), VideoMind demonstrates its effectiveness on both key event grounding and question answering.

**Video Temporal Grounding** We also evaluate the grounder and verifier on video temporal grounding datasets. The results on Charades-STA (Gao et al., 2017) and ActivityNet-Captions (Krishna et al., 2017) are shown in Table 5. Benefiting from (1) the timestamp decoder design, and (2) a verifier that refines the results by focusing on critical segments, our model surpasses all LLM-based temporal grounding methods and yields competitive results compared to fine-tuned experts.

**General Video Question Answering** We are also interested in whether our temporally augmented design can improve general VideoQA tasks. In Table 6, we evaluate our model on three long video benchmarks to determine if the Chain-of-LoRA design generalizes to common settings. Our designs effectively help the model localize cue segments before answering the question.

### 4.2 Q2: THE ADVANTAGES OF CHAIN-OF-LORA

Table 7 studies the effect of role integration on VideoMind-2B. First, text-based CoT does not improve the base model, highlighting the need for a vision-centric reasoning strategy. Second, the key capabilities of roles may conflict with one another, thus only sub-optimal performance can be

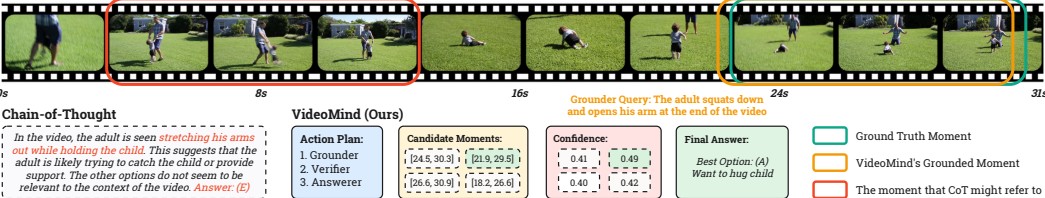

Figure 6: Visualization of the reasoning process of VideoMind. Through chaining the `planner`, `grounder`, `verifier`, and `answerer`, our model accurately localizes the critical moment and selects the correct answer, avoiding confusion from incorrect segments.

Table 7: Performance and efficiency comparison of different test-time scaling and role integration strategies. Mem indicates the peak GPU memory consumption. Notably, Chain-of-LoRA achieves the best performance with minimal memory cost.

| Method | Mem | NExT-GQA | | Charades-STA | | Video-MME | |
|---|---|---|---|---|---|---|---|
| | | mIoU | Acc | R@0.5 | mIoU | All | Long |
| Qwen2-VL-2B | 4.1G | – | 69.6 | – | – | 53.0 | 43.1 |
| +CoT | 4.1G | – | 69.7 | – | – | 52.8 | 43.3 |
| +All-in-One | 4.2G | 28.0 | 70.5 | 47.8 | 42.1 | 53.6 | 43.6 |
| +All-Distributed | 16.6G | 28.6 | 71.4 | 51.1 | 45.2 | 55.4 | 46.3 |
| **+Chain-of-LoRA** | **4.2G** | **28.6** | **71.4** | **51.1** | **45.2** | **55.4** | **46.3** |

Table 8: Effects of individual roles. A, G, V, P, G% denote the answerer, grounder, verifier, planner, and the percentage of samples processed with the grounder, respectively.

| Roles To Use | | | | | ReXTime | | Charades-STA | | |
|---|---|---|---|---|---|---|---|---|---|
| A | G | V | P | G% | mIoU | Acc | R@0.5 | R@0.7 | mIoU |
| ✓ | | | | 0% | – | 68.0 | – | – | – |
| ✓ | ✓ | | | 100% | 24.5 | 68.8 | – | – | – |
| ✓ | ✓ | ✓ | | 100% | 24.8 | 69.1 | – | – | – |
| ✓ | ✓ | ✓ | ✓ | 100% | 24.7 | 69.2 | – | – | – |
| ✓ | ✓ | ✓ | ✓ | 40% | **26.7** | **70.0** | – | – | – |
| ✓ | | | | | – | – | 47.2 | 21.7 | 42.0 |
| ✓ | ✓ | | | | – | – | **51.1** | **26.0** | **45.2** |

achieved via joint training. Compared to the all-distributed approach that requires multiple copies (4×) of weights, Chain-of-LoRA offers the best balance between effectiveness and efficiency.

### 4.3 Q3: KEY ABLATION STUDIES

**Effect of Individual Roles** The contributions of different roles are studied in Table 8. Our observations are as follows: (1) `Grounder:` By identifying visual cues, the grounder can slightly improve QA accuracy, indicating that the grounder is especially effective on long videos. (2) `Verifier:` Selecting the best candidate through the verifier improves grounding performance, yielding a consistent gain of 3.2 mIoU on Charades-STA. (3) `Planner:` Coordinating roles via the planner – even when performing grounding on only 40% samples (the remaining 60% are directly processed by the answerer) – boosts the accuracy from 69.2 to 70.0. This highlights the model's flexibility to adaptively determine whether to perform grounding under different temporal contexts.

### 4.4 VISUALIZATION

In Figure 6, we illustrate how VideoMind applies all roles to progressively derive the correct answer while avoiding potential mistakes. The planner determines what roles are needed, then calls the grounder to generate candidate moments. The verifier selects the most relevant segment (highlighted in yellow), which is then zoomed-in and passed to the answerer for further reasoning.

## 5 CONCLUSION

In this work, we introduced **VideoMind**, a video-language agent designed for temporal-grounded video reasoning. Our approach employs an agentic workflow consisting of four carefully designed roles along with a **Chain-of-LoRA** strategy to flexibly switch among them. Extensive experiments on Grounded VideoQA, Video Temporal Grounding, and General VideoQA tasks demonstrate the effectiveness and significance of our method, particularly in long-form video reasoning by providing evidence-based answers. We hope this work inspires future advancements in agentic reasoning.

**Limitations & Future Work** We acknowledge that our method requires careful optimization of individual designs and preparation of training data. In our future work, we will investigate (1) the possibility of joint-optimization of multiple roles and (2) the integration of audio modality.

## ACKNOWLEDGEMENTS

This research is supported by The Hong Kong Research Grants Council (GRF-15229423) and by Tencent's research donation. We also acknowledge The University Research Facility in Big Data Analytics (UBDA) at The Hong Kong Polytechnic University for providing computing resources that have contributed to the research results reported within this paper. Mike Shou does not receive any funding for this work.

## ETHICS STATEMENT

This study focuses on algorithmic innovations for improving the visual reasoning capabilities of large multi-modal models. It does not involve human subjects, private data, or any potentially harmful insights. All datasets used are publicly available and widely adopted in the community. We acknowledge the potential risks of misuse associated with LLMs and LMMs, including the bias propagation and harmful content generation. However, this study does not directly address the deployment or generation. Instead, it contributes to the understanding of the model architecture and the reasoning mechanism. To the best of our knowledge, our research complies with the ICLR Code of Ethics and does not involve any known violations or harms.

## REPRODUCIBILITY STATEMENT

We are committed to ensuring the full reproducibility of this study. To achieve this, we have provided key hyperparameters settings in Section 3, formulation of inference pipeline in Section A.1, implementation details in Section A.2, evaluation metrics in Section B.1, and prompt templates in Section C.1. We also open-source all the code, model checkpoints, data, and training logs in this study to facilitate future research in this direction.

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

## APPENDIX

In this appendix, we provide more details about the model inference pipeline and implementation details to complement the main paper. Additional experiments, detailed analysis, and discussions are also incorporated. Below is the table of contents.

## A  MODEL

### A.1  INFERENCE PIPELINE

The formulation of VideoMind's inference pipeline is illustrated in Algorithm 1. Given a video $\mathcal{V}$ and a question $\mathcal{Q}$, the planner dynamically calls different roles on demand to analyze the multimodal context and generate the answer.

---

**Algorithm 1** VideoMind's Chain-of-LoRA Pipeline

---

1: **Input:** A video $\mathcal{V}$ and a question $\mathcal{Q}$
2: **Output:** An answer $\mathcal{A}$ to the question with temporal moment $\mathcal{T} = [t_s, t_e]$
3: Plan $\mathcal{P} \leftarrow$ **Planner**$(\mathcal{V}, \mathcal{Q})$
4: **if Grounder** $\in \mathcal{P}$ **then**
5:     $\{[t_s^i, t_e^i]\}_i \leftarrow$ **Grounder**$(\mathcal{V}, \mathcal{Q})$
6:     **for all** $i$ **do**
7:         $\tilde{\mathcal{V}}_i \leftarrow$ ZoomIn$(\mathcal{V}, [t_s^i, t_e^i])$
8:         $Score_i \leftarrow$ **Verifier**$(\tilde{\mathcal{V}}_i, \mathcal{Q})$
9:     **end for**
10:    $i \leftarrow \arg\max s_i(Score_i)$
11: **end if**
12: **if Answerer** $\in \mathcal{P}$ **then**
13:    $\mathcal{A} \leftarrow$**Answerer**$(\tilde{\mathcal{V}}_i, \mathcal{Q})$
14: **end if**
15: **return** $(\mathcal{A}, \mathcal{T})$

---

### A.2  IMPLEMENTATION DETAILS

We leverage the 2B and 7B versions of Qwen2-VL (Wang et al., 2024b) as our base models, and apply LoRA adapters with $\mathrm{rank} = 64$ and $\mathrm{alpha} = 64$ to the planner, grounder, and verifier. The hidden size of the timestamp decoder is set to 256. The maximum number of tokens per frame and maximum number of frames for the planner, grounder, verifier, and answerer are set as [64, 100], [64, 150], [64, 64], and [256, 32], respectively. We train different roles separately on different datasets and load them together during inference, so that the model can efficiently switch roles by activating different LoRAs. During training, we set the global batch size to 32, and utilize the AdamW optimizer (Loshchilov & Hutter, 2019) with learning rates of 2e-5, 1e-4, and 5e-5 for planner, grounder, and verifier, respectively. All the roles were trained for 1 epoch on their specific datasets, with a linear warmup in the first 3% steps. During inference, we apply an NMS with $\mathrm{IoU} = 0.75$ to reduce duplicated moments from the grounder.

Table 9: Details of the evaluation benchmarks. The datasets encompass three representative tasks, *i.e.*, Grounded VideoQA, Video Temporal Grounding, and General VideoQA, with video durations ranging from several seconds to more than one hour.

| Dataset | Duration | Domain | Main Metrics |
|---------|----------|--------|--------------|
| *Grounded Video Question Answering (Grounding + QA)* | | | |
| CG-Bench (Chen et al., 2024a) | 1624.4s | Diverse | rec.@IoU, acc.@IoU |
| ReXTime (Chen et al., 2024c) | 141.1s | Vlog, News, Activity | mIoU, Acc (IoU ⩾ 0.5) |
| NExT-GQA (Xiao et al., 2024) | 39.5s | Reasoning | mIoP, Acc@GQA |
| *Video Temporal Grounding (Grounding only)* | | | |
| Charades-STA (Gao et al., 2017) | 30.1s | Indoor | R@{0.3 ∼ 0.7}, mIoU |
| ActivityNet-Captions (Krishna et al., 2017) | 111.4s | Activity | R@{0.3 ∼ 0.7}, mIoU |
| QVHighlights (Lei et al., 2021) | 150s | Vlog, News | R@{0.5, 0.7}, mAP |
| TACoS (Regneri et al., 2013) | 358.2s | Cooking | R@{0.3 ∼ 0.7}, mIoU |
| Ego4D-NLQ (Grauman et al., 2022) | 379.0s | Egocentric | R@{0.3 ∼ 0.7}, mIoU |
| ActivityNet-RTL (Huang et al., 2024b) | 111.4s | Reasoning | P@0.5, mIoU |
| *General Video Question Answering (QA only)* | | | |
| Video-MME (Fu et al., 2024a) | 1017.9s | Diverse | Acc (*w/o subs*) |
| MLVU (Zhou et al., 2024) | 930s | Diverse | Acc |
| LVBench (Wang et al., 2024c) | 4101s | Diverse | Acc |
| MVBench (Li et al., 2024b) | 15s | Diverse | Acc |
| LongVideoBench (Wu et al., 2024) | 473s | Diverse | Acc |

Table 10: Performance on MultiHop-EgoQA (Chen et al., 2025a). FT means fine-tuning on the target dataset. Sent. Sim. denotes sentence similarity computed by `all-MiniLM-L6-v2`.

| Method | Size | FT | Temporal Grounding | | Question Answering | |
|--------|------|----|--------------------|--|--------------------|--|
| | | | IoU@0.3 | mIoU | Sent. Sim. | Score |
| Human | – | – | 87.0 | 61.8 | 74.3 | 7.5 |
| GPT-4o (OpenAI, 2024a) | – | ✗ | 12.0 | 12.2 | 73.7 | **5.4** |
| InternVL2 (OpenGVLab, 2024) | 8B | ✗ | 6.3 | 6.6 | 71.9 | 4.5 |
| LLaVA-NeXT-Video (Liu et al., 2024a) | 7B | ✗ | – | – | 62.1 | 4.2 |
| TimeChat (Ren et al., 2024) | 7B | ✗ | 3.0 | 3.6 | 58.9 | 3.3 |
| VTimeLLM (Huang et al., 2024a) | 7B | ✗ | 8.8 | 9.2 | 70.5 | 4.3 |
| GeLM (Chen et al., 2025a) | 7B | ✓ | 18.2 | 16.7 | 75.0 | 4.8 |
| **VideoMind** (Ours) | 2B | ✗ | 23.2 | 17.8 | 58.8 | 3.5 |
| **VideoMind** (Ours) | 7B | ✗ | **25.1** | **19.0** | **77.3** | 4.9 |

# B  EXPERIMENTS

## B.1  BENCHMARKS AND SETTINGS

The experiments are extensively designed across 15 diverse benchmarks. The statistics are listed in Table 9. The major benchmarks are introduced below.

**CG-Bench** (Chen et al., 2024a) is designed for long video grounded question answering, featuring a diverse domain and various evaluation metrics. It includes 1.2K manually curated videos, ranging from 10 to 80 minutes, with a total of 12K QA pairs. The dataset is categorized into perception, reasoning, and hallucination question types, and introduces clue-based evaluation methods like white box and black box assessments to ensure models provide answers based on accurate video reasoning.

**ReXTime** (Chen et al., 2024c) tests models on complex temporal reasoning, using an automated pipeline for QA pair generation, significantly reducing manual effort. It includes 921 validation and 2.1K test samples, each manually curated for accuracy, and highlights a 14.3% accuracy gap between SoTA models and human performance. This benchmark is crucial for evaluating models on cause-and-effect relationships across video segments.

**NExT-GQA** (Xiao et al., 2024) aims to challenge models to reason about causal and temporal actions, supporting both multi-choice and open-ended tasks. This is an extension of NExT-QA (Xiao et al., 2021) comprising 10.5K manually labeled video QA pairs with temporal segments. The samples in this benchmark are from "causal" and "temporal" classes, while the "descriptive" questions in NExT-QA are discarded.

Table 11: Video temporal grounding on TACoS (Regneri et al., 2013). FT means fine-tuning on the target dataset. Note that our method was co-trained on this dataset.

| Method | Size | FT | R@0.3 | R@0.5 | R@0.7 | mIoU |
|---|---|---|---|---|---|---|
| *Non-LLM-based Specialists* | | | | | | |
| 2D-TAN (Zhang et al., 2020b) | – | ✓ | 40.0 | 28.0 | 12.9 | 27.2 |
| Moment-DETR (Lei et al., 2021) | – | ✓ | 38.0 | 24.7 | 12.0 | 25.5 |
| UniVTG (Lin et al., 2023b) | – | ✓ | 51.4 | 35.0 | 17.4 | 33.6 |
| $R^2$-Tuning (Liu et al., 2024d) | – | ✓ | 49.7 | 38.7 | 25.1 | 35.9 |
| *LLM-based Generalists* | | | | | | |
| **VideoMind** (Ours) | 2B | ✗ | 38.6 | 26.9 | 15.5 | 27.4 |
| **VideoMind** (Ours) | 7B | ✗ | **49.5** | **36.2** | **21.4** | **34.4** |

Table 12: Performance of video temporal grounding on Ego4D-NLQ (Grauman et al., 2022). FT means fine-tuning on the target dataset. VideoMind-Ego is a variant of our method trained with extra 67K egocentric grounding samples from NaQ (Ramakrishnan et al., 2023).

| Method | Size | FT | R@0.3 | R@0.5 | R@0.7 | mIoU |
|---|---|---|---|---|---|---|
| *Non-LLM-based Specialists* | | | | | | |
| 2D-TAN (Zhang et al., 2020b) | – | ✓ | 4.3 | 1.8 | 0.6 | 3.4 |
| VSLNet (Zhang et al., 2020a) | – | ✓ | 4.5 | 2.4 | 1.0 | 3.5 |
| Moment-DETR (Lei et al., 2021) | – | ✓ | 4.3 | 1.8 | 0.7 | 3.5 |
| UniVTG (Lin et al., 2023b) | – | ✓ | 7.3 | 4.0 | 1.3 | 4.9 |
| $R^2$-Tuning (Liu et al., 2024d) | – | ✓ | 7.2 | 4.5 | 2.1 | 4.9 |
| UniVTG (Lin et al., 2023b) | – | ✗ | 6.5 | 3.5 | 1.2 | 4.6 |
| *LLM-based Generalists* | | | | | | |
| **VideoMind** (Ours) | 2B | ✗ | 5.9 | 2.9 | 1.2 | 4.7 |
| **VideoMind** (Ours) | 7B | ✗ | **7.2** | 3.7 | 1.7 | **5.4** |
| **VideoMind-Ego** (Ours) | 2B | ✗ | **7.2** | **3.9** | **1.8** | 5.3 |

**Charades-STA (Gao et al., 2017)** contains 10K in-door videos, averaging 30.1 seconds each, with 16K temporal annotations spanning daily activity, alongside free-text descriptions. These rich annotations make Charades-STA particularly suitable for evaluating temporal grounding models under indoor environments.

**ActivityNet-Captions (Krishna et al., 2017)** is a large-scale benchmark with 20K untrimmed YouTube videos with a total of 849 hours, covering diverse activities from personal care to sports. This dataset contains high-quality dense video captioning annotations (3.65 temporally localized sentences per video), which we use as queries for video temporal grounding. Each query has an average length of 13.5 words.

## B.2 More Experimental Results

**Multi-Hop Grounded Question Answering** To investigate the performance of our method on novel tasks that require a hybrid or dynamically generated sequence of steps, we evaluate our method on MultiHop-EgoQA (Chen et al., 2025a), a Grounded VideoQA dataset highlighting multi-hop temporal reasoning. For each question, the model must ground and reason on multiple relevant moments before answering, which is a paradigm that does not fit neatly into the pre-defined single-hop grounding pipeline. The evaluation results are shown in Table 10. Thanks to VideoMind's architectural design to produce multiple candidate moments in a single grounding step, it can effectively capture the multi-hop evidence required by this benchmark. As a result, our method achieves strong zero-shot performance, surpassing all open-source baselines and remaining competitive to closed-source GPT-4o (OpenAI, 2024a) across both grounding metrics and QA metrics.

**Video Temporal Grounding** We additionally compare VideoMind with representative methods on the challenging TACoS (Regneri et al., 2013), Ego4D-NLQ (Grauman et al., 2022), and QVHighlights (Lei et al., 2021) datasets in Table 11, Table 12, and Table 13, respectively. Our 2B model performs better than the strong task-specific baseline UniVTG (Lin et al., 2023b) on TACoS but slightly worse than it on Ego4D-NLQ. This is justifiable as neither the grounder nor the verifier was

Table 13: Fine-tuned video temporal grounding results on QVHighlights (Lei et al., 2021).

| Method | Size | R1 @0.5 | R1 @0.7 | mAP @0.5 | mAP @0.75 | Avg. |
|---|---|---|---|---|---|---|
| *Non-LLM-based Specialists* | | | | | | |
| XML (Lei et al., 2020) | – | 41.83 | 30.35 | 44.63 | 31.73 | 32.14 |
| XML+ (Lei et al., 2021) | – | 46.69 | 33.46 | 47.89 | 34.67 | 34.90 |
| Moment-DETR (Lei et al., 2021) | – | 59.78 | 40.33 | 60.51 | 35.36 | 36.14 |
| UMT (Liu et al., 2022) | – | 60.83 | 43.26 | 57.33 | 39.12 | 38.08 |
| MomentDiff (Li et al., 2023b) | – | 58.21 | 41.48 | 54.57 | 37.21 | 36.84 |
| QD-DETR (Moon et al., 2023) | – | 62.40 | 44.98 | 62.52 | 39.88 | 39.86 |
| UniVTG (Lin et al., 2023b) | – | 65.43 | 50.06 | 64.06 | 45.02 | 43.63 |
| R²-Tuning (Liu et al., 2024d) | – | 68.03 | 49.35 | 69.04 | 47.56 | 46.17 |
| *LLM-based Generalists* | | | | | | |
| **VideoMind** (Ours) | 2B | 75.42 | 59.35 | 74.11 | 55.15 | 51.60 |
| **VideoMind** (Ours) | 7B | 78.53 | 61.09 | 76.07 | 58.17 | 54.19 |

Table 14: Comparison of performance on reasoning temporal localization on ActivityNet-RTL (Huang et al., 2024b). Our zero-shot VideoMind-7B outperforms the strong fine-tuned baseline LITA-13B (Huang et al., 2024b) by a considerable margin.

| Method | Size | FT | P@0.5 | mIoU |
|---|---|---|---|---|
| LITA (Huang et al., 2024b) | 7B | ✓ | 21.2 | 24.1 |
| LITA (Huang et al., 2024b) | 13B | ✓ | 25.9 | 28.6 |
| **VideoMind** (Ours) | 2B | ✗ | 20.1 | 22.7 |
| **VideoMind** (Ours) | 7B | ✗ | 28.0 | 31.3 |

Table 15: Performance of VideoQA on LongVideoBench (Wu et al., 2024) `val` split.

| Method | Size | Acc | Acc @ Duration Groups (8, 15] | (15, 60] | (180, 600] | (900, 3600] |
|---|---|---|---|---|---|---|
| GPT-4o (OpenAI, 2024a) | – | 66.7 | 71.4 | 76.7 | 69.1 | 60.9 |
| GPT-4 Turbo (Achiam et al., 2023) | – | 59.0 | 65.2 | 68.2 | 62.4 | 50.5 |
| Gemini-1.5-Pro (Reid et al., 2024) | – | 64.0 | 67.4 | 75.1 | 65.3 | 58.6 |
| Gemini-1.5-Flash (Reid et al., 2024) | – | 61.6 | 68.3 | 76.2 | 62.6 | 54.0 |
| Idefics2 (Laurencon et al., 2024) | 8B | 49.7 | 59.8 | 65.7 | 47.8 | 42.7 |
| Phi-3-Vision (Abdin et al., 2024) | 4B | 49.6 | 59.3 | 61.6 | 46.8 | 44.7 |
| Mantis-Idefics2 (Jiang et al., 2024) | 8B | 47.0 | 56.6 | 55.8 | 45.6 | 42.2 |
| Mantis-BakLLaVA (Jiang et al., 2024) | 7B | 43.7 | 53.4 | 57.6 | 40.3 | 38.7 |
| **VideoMind** (Ours) | 2B | 48.8 | 59.3 | 59.3 | 49.3 | 41.7 |
| **VideoMind** (Ours) | 7B | 56.3 | 67.7 | 67.4 | 56.8 | 48.6 |

Table 16: Performance comparison on general VideoQA on MVBench (Li et al., 2024b).

| Model | Size | AS | AP | AA | FA | UA | OE | OI | OS | MD | AL | ST | AC | MC | MA | SC | FP | CO | EN | ER | CI | Avg. |
|---|---|---|---|---|---|---|---|---|---|---|---|---|---|---|---|---|---|---|---|---|---|---|
| GPT-4V (OpenAI, 2023) | – | 55.5 | 63.5 | 72.0 | 46.5 | 73.5 | 18.5 | 59.0 | 29.5 | 12.0 | 40.5 | 83.5 | 39.0 | 12.0 | 22.5 | 45.0 | 47.5 | 52.0 | 31.0 | 59.0 | 11.0 | 43.5 |
| Video-ChatGPT (Maaz et al., 2023) | 7B | 23.5 | 26.0 | 62.0 | 22.5 | 26.5 | 54.0 | 28.0 | 40.0 | 23.0 | 20.0 | 31.0 | 30.5 | 25.5 | 39.5 | 48.5 | 29.0 | 33.0 | 29.5 | 26.0 | 35.5 | 32.7 |
| Video-LLaMA (Zhang et al., 2023b) | 7B | 27.5 | 25.5 | 51.0 | 29.0 | 39.0 | 48.0 | 40.5 | 38.0 | 22.5 | 22.5 | 43.0 | 34.0 | 22.5 | 32.5 | 45.5 | 32.5 | 40.0 | 30.0 | 21.0 | 37.0 | 34.1 |
| VideoChat (Li et al., 2023a) | 7B | 33.5 | 26.5 | 56.0 | 33.5 | 40.5 | 53.0 | 40.5 | 30.0 | 25.5 | 27.0 | 48.5 | 35.0 | 20.5 | 42.5 | 46.0 | 26.5 | 41.0 | 23.5 | 23.5 | 36.0 | 35.5 |
| Video-LLaVA (Lin et al., 2023a) | 7B | 46.0 | 42.5 | 56.5 | 39.0 | 53.5 | 53.0 | 48.0 | 41.0 | 29.0 | 31.5 | 82.5 | 45.0 | 26.0 | 53.0 | 41.5 | 33.5 | 41.5 | 27.5 | 38.5 | 31.5 | 43.0 |
| TimeChat (Ren et al., 2024) | 7B | 40.5 | 36.0 | 61.0 | 32.5 | 53.0 | 53.5 | 41.5 | 29.0 | 19.5 | 26.5 | 66.5 | 34.0 | 20.0 | 43.5 | 42.0 | 36.5 | 36.0 | 29.0 | 35.0 | 35.0 | 38.5 |
| PLLaVA (Xu et al., 2024) | 7B | 58.0 | 49.0 | 55.5 | 41.0 | 61.0 | 56.0 | 61.0 | 36.0 | 23.5 | 26.0 | 82.0 | 39.5 | 42.0 | 52.0 | 45.0 | 42.0 | 53.5 | 30.5 | 48.0 | 31.0 | 46.6 |
| ShareGPT4Video (Chen et al., 2024d) | 7B | 49.5 | 39.5 | 79.5 | 40.0 | 54.5 | 82.5 | 54.5 | 32.5 | 50.5 | 41.5 | 84.5 | 35.5 | 62.5 | 75.0 | 51.0 | 25.5 | 46.5 | 28.5 | 39.0 | 51.5 | 51.2 |
| ST-LLM (Liu et al., 2024c) | 7B | 66.0 | 53.5 | 84.0 | 44.0 | 58.5 | 80.5 | 73.5 | 38.5 | 42.5 | 31.0 | 86.5 | 36.5 | 56.5 | 78.5 | 43.0 | 44.5 | 46.5 | 34.5 | 41.5 | 58.5 | 54.9 |
| VideoGPT+ (Maaz et al., 2024) | 3.8B | 69.0 | 60.0 | 83.0 | 48.5 | 66.5 | 85.5 | 75.5 | 36.0 | 44.0 | 34.0 | 89.5 | 39.5 | 71.0 | 90.5 | 45.0 | 53.0 | 50.0 | 29.5 | 44.0 | 60.0 | 58.7 |
| VideoChat2 (Li et al., 2024b) | 7B | 75.5 | 58.0 | 83.5 | 50.5 | 60.5 | 87.5 | 74.5 | 45.0 | 47.5 | 44.0 | 82.5 | 37.0 | 64.5 | 87.5 | 51.0 | 66.5 | 47.0 | 35.0 | 37.0 | 72.5 | 60.4 |
| **VideoMind** (Ours) | 2B | 78.5 | 76.0 | 75.5 | 46.0 | 69.5 | 90.5 | 71.5 | 33.0 | 48.0 | 40.0 | 92.5 | 52.5 | 71.5 | 92.0 | 46.5 | 61.5 | 61.5 | 37.5 | 51.0 | 57.0 | 62.5 |
| **VideoMind** (Ours) | 7B | 74.0 | 71.5 | 81.0 | 50.0 | 77.0 | 93.0 | 75.0 | 38.0 | 48.5 | 46.0 | 91.0 | 39.0 | 80.0 | 94.5 | 49.5 | 55.5 | 70.0 | 40.5 | 57.0 | 61.0 | 64.6 |

trained on egocentric videos, while UniVTG was pretrained on 1.8M samples from Ego4D (Grauman et al., 2022). To align the setting, we trained an additional VideoMind-2B variant with extra 67K grounding samples from NaQ (Ramakrishnan et al., 2023). To our best knowledge, VideoMind is **the first LLM-based grounding model that supports multi-moment outputs**, thereby being able to be evaluated on QVHighlights. Compared with task-specific experts, our VideoMind-2B significantly outperforms all previous methods and sets a new state-of-the-art.

**Reasoning Temporal Localization** We also evaluate the generalizability of grounder and verifier on the more challenging reasoning temporal localization (Huang et al., 2024b) task, which is similar to video temporal grounding, but the queries are not directly describing the moment. The models are required to infer the actual event using their world knowledge. The results in Table 14 show that VideoMind can successfully generalize its zero-shot grounding capability to complex scenarios.

**General Video Question Answering** For the task of long VideoQA, we also provide evaluations on LongVideoBench (Wu et al., 2024) in Table 15, which further verifies the effectiveness of Video-

Table 17: Comparison with representative video reasoning methods on video QA/grounding tasks.

| Method | Size | CG-Bench | MLVU | LVBench | Charades-STA | | ActivityNet-Captions | |
|---|---|---|---|---|---|---|---|---|
| | | long-acc. | M-Avg | Overall | R@0.5 | mIoU | R@0.5 | mIoU |
| *Pure Text-based Reasoning Models* | | | | | | | | |
| LongVILA-R1 (Chen et al., 2025b) | 7B | 26.7 | 56.5 | 34.7 | 30.3 | 30.0 | 16.4 | 21.4 |
| Video-R1 (Feng et al., 2025) | 7B | 34.4 | 63.1 | 38.4 | 35.3 | 34.9 | 22.6 | 28.0 |
| *Vision-centric Reasoning Models* | | | | | | | | |
| **VideoMind** (Ours) | 7B | **38.4** | **64.4** | **40.8** | **59.1** | **50.2** | **30.3** | **33.3** |

Table 18: Performance of different timestamp modeling designs on Charades-STA (Gao et al., 2017).

| Method | R@0.3 | R@0.5 | R@0.7 | mIoU |
|---|---|---|---|---|
| Text-only (Ren et al., 2024) | 56.8 | 39.5 | 14.3 | 36.1 |
| Special Tokens (Qian et al., 2024a) | 56.4 | 39.2 | 14.5 | 35.7 |
| Embedding Matching (Liu et al., 2024e) | 59.6 | 43.5 | 17.0 | 38.2 |
| Time Marker (Chen et al., 2024e) | 60.5 | 43.9 | 17.2 | 38.6 |
| **Timestamp Decoder** (Ours) | **64.1** | **47.2** | **21.7** | **42.0** |

Table 19: Case distribution on ReXTime (Chen et al., 2024c) and NExT-GQA (Xiao et al., 2024). *Correct*, *Planning*, *Grounding*, *Verification*, and *Answering* refers to correct prediction, planning error, grounding error, verification error, and answering error, respectively.

| Method | Size | ReXTime | | | | | NExT-GQA | | | | |
|---|---|---|---|---|---|---|---|---|---|---|---|
| | | Correct | Planning | Grounding | Verification | Answering | Correct | Planning | Grounding | Verification | Answering |
| **VideoMind** | 2B | 69.1% | 1.2% | 18.3% | 5.7% | 5.7% | 71.2% | 1.9% | 14.0% | 6.9% | 6.0% |
| **VideoMind** | 7B | 74.6% | 1.1% | 15.0% | 4.6% | 4.7% | 76.6% | 0.7% | 11.8% | 5.8% | 5.1% |

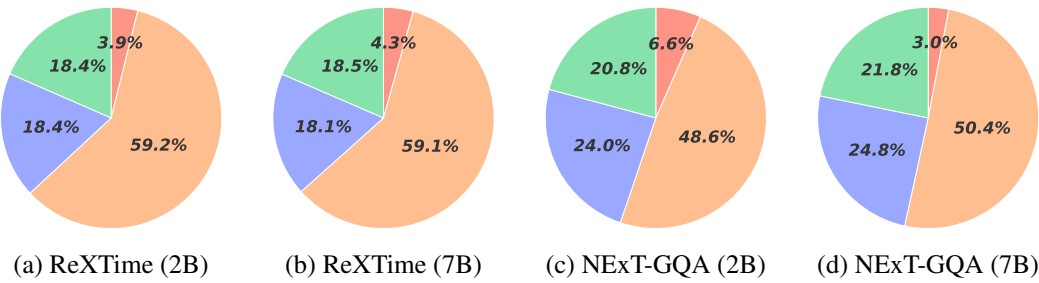

(a) ReXTime (2B)     (b) ReXTime (7B)     (c) NExT-GQA (2B)     (d) NExT-GQA (7B)

Figure 7: Error distribution of our 2B and 7B variants on ReXTime (Chen et al., 2024c) and NExT-GQA (Xiao et al., 2024) datasets. The red, orange, blue, and green portions represent planning, grounding, verification, and answering errors, respectively.

**Mind on videos scaling to one-hour long.** Table 16 presents more results of VideoMind on MVBench (Li et al., 2024b), which is a benchmark with very short videos (around 15s). Our model can still achieve good performance on these short video scenarios.

**Comparison with Text-based Reasoning Models** In Table 17, we compare our method with representative pure text-based video reasoning methods. Our method significantly outperforms both baselines on all benchmarks, demonstrating that vision-centric reasoning is superior to pure text-based reasoning on long/complex video reasoning tasks.

## B.3 MORE DETAILED ANALYSIS

**Timestamp Modeling Designs** The grounder plays a crucial role in our proposed Chain-of-LoRA pipeline. The model's temporal grounding quality directly impacts the final QA accuracy. To demonstrate the necessity of this design, we implement and compare the following alternative timestamp modeling techniques based on VideoMind-2B (Grounder):

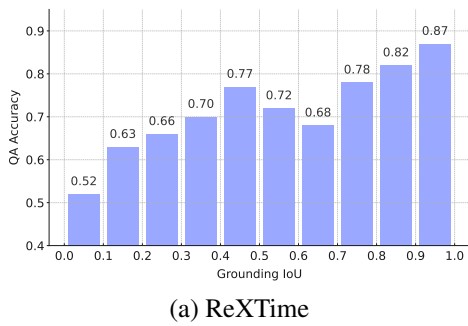
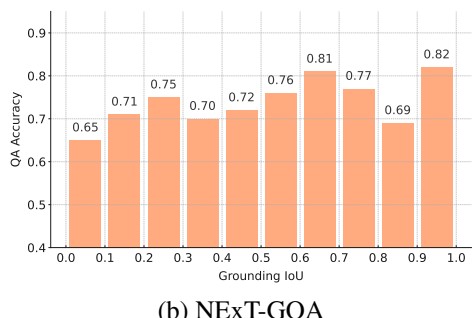

(a) ReXTime

(b) NExT-GQA

Figure 8: The correlation between grounding IoU and the final QA accuracy of VideoMind-2B on ReXTime (Chen et al., 2024c) and NExT-GQA (Xiao et al., 2024) datasets.

Table 20: Effect of the temporal feature pyramid on Charades-STA (Gao et al., 2017).

| #Pyramid Levels | Charades-STA | | | |
|---|---|---|---|---|
| | R@0.3 | R@0.5 | R@0.7 | mIoU |
| 1 | 60.55 | 44.57 | 15.82 | 38.13 |
| 2 | 61.51 | 46.90 | 19.36 | 40.43 |
| 3 | 62.62 | 47.02 | 20.08 | 41.27 |
| 4 | **63.55** | **47.23** | **21.69** | **42.02** |

Table 21: Effect of different verifier styles on Charades-STA (Gao et al., 2017).

| Verifier Type | Charades-STA | | | |
|---|---|---|---|---|
| | R@0.3 | R@0.5 | R@0.7 | mIoU |
| Direct | 60.42 | 45.28 | 19.32 | 39.84 |
| Expand | 65.10 | 48.70 | 23.15 | 43.57 |
| Textual | 65.24 | 49.33 | 23.89 | 44.01 |
| Special Token | **67.63** | **51.05** | **25.99** | **45.22** |

Table 22: Effect of the verifier on Charades-STA (Gao et al., 2017). IoU Raise means the percentage of the samples whose grounding IoU is raised by the verifier.

| Role(s) | Size | R@0.3 | R@0.5 | R@0.7 | mIoU | IoU Raise |
|---|---|---|---|---|---|---|
| Grounder | 2B | 63.2 | 46.9 | 20.5 | 41.7 | – |
| Grounder + Verifier | 2B | **68.0** (+7.6%) | **51.2** (+9.2%) | **24.3** (+18.5%) | **44.8** (+7.4%) | 32.9% |
| Grounder | 7B | 69.4 | 53.2 | 26.6 | 46.8 | – |
| Grounder + Verifier | 7B | **73.8** (+6.3%) | **59.1** (+11.1%) | **30.1** (+13.2%) | **49.8** (+6.4%) | 31.3% |

Table 23: The accuracy of planner with different input combinations.

| Input Video | Input Question | Planning Acc |
|---|---|---|
| ✓ | | 0.42 |
| | ✓ | 0.79 |
| ✓ | ✓ | **0.93** |

Table 24: Comparison of average inference time on CG-Bench (Chen et al., 2024a) (avg. duration: 27 min).

| Method | Size | Inference Time (s/video) |
|---|---|---|
| LongVILA-R1 (Chen et al., 2025b) | 7B | 8.75 |
| **VideoMind** | 7B | 9.53 (+8.9%) |
| **VideoMind** (w. Auto Planning) | 7B | **8.07** (-7.8%) |

1. **Text-only (Ren et al., 2024):** Directly represent timestamps in text form (*e.g.*, "2.3 seconds").
2. **Special Tokens (Qian et al., 2024a):** Define a set of timestamp tokens (*e.g.*, `<T0>`, `<T1>`).
3. **Embedding Matching (Liu et al., 2024e):** Predict frame features to retrieve the frame index.
4. **Time Marker (Chen et al., 2024e):** Explicitly insert textual timestamps among visual tokens.

Their zero-shot video temporal grounding results are shown in Table 18. The results clearly demonstrate that the timestamp decoder delivers the strongest temporal grounding capability. We attribute it to two key factors: (1) It decouples continuous timestamp modeling from discrete token prediction, allowing the model to represent time with higher precision; (2) The direct regression supervision (L1 Loss) further enhances time reasoning and stabilizes training. Moreover, the timestamp decoder naturally supports predicting multiple moments with corresponding confidence scores, supporting tasks like multi-moments retrieval (Lei et al., 2021) and facilitating moment re-ranking through the verifier. These advantages jointly enhance the reliability of temporal grounding, which ensures the correct moment could be localized for further reasoning.

**Effect of the Temporal Feature Pyramid** Table 20 studies the effectiveness of the temporal feature pyramid. Our baseline model directly makes predictions on the last-layer transformer outputs. When adding more pyramid levels, the performance of video temporal grounding consistently im-

Table 25: Controlled experiments with strictly aligned hyperparameter settings. Both MLVU (Zhou et al., 2024) and LVBench (Wang et al., 2024c) are downsampled to 300 samples each.

| Method | Size | MLVU (mini) | LVBench (mini) |
|---|---|---|---|
| | | M-Avg | Overall |
| GPT-4o (OpenAI, 2024a) | – | 59.7 | 31.3 |
| Gemini-1.5-Pro (Reid et al., 2024) | – | 60.3 | 36.3 |
| VideoMind (Ours) | 2B | 59.3 | 35.7 |
| VideoMind (Ours) | 7B | 62.7 | 40.3 |

Table 26: Performance comparison among the integration of our Chain-of-LoRA mechanism on different representative base models.

| Base Model | Size | CG-Bench | ReXTime | Video-MME | MLVU | LVBench |
|---|---|---|---|---|---|---|
| | | acc.@IoU | Acc@IoU | w/o sub. | M-Avg | Overall |
| Qwen2-VL (Wang et al., 2024b) | 2B | 4.0 | 17.3 | 55.4 | 58.7 | 35.4 |
| | 7B | 4.7 | 20.2 | 58.2 | 64.4 | 40.8 |
| Qwen2.5-VL (Bai et al., 2025) | 3B | 5.0 | 15.6 | 60.9 | 62.7 | 40.5 |
| | 7B | 5.7 | 19.8 | 65.9 | 66.3 | 45.2 |
| InternVL3 (Zhu et al., 2025) | 2B | 4.1 | 17.5 | 58.2 | 61.4 | 38.1 |
| | 8B | 4.5 | 20.8 | 66.5 | 63.8 | 42.3 |

Table 27: Performance of the simulated multi-role pipelines on closed-source models. Both MLVU (Zhou et al., 2024) and LVBench (Wang et al., 2024c) are downsampled to 300 samples each.

| Method | Multi-Role Pipeline | MLVU (mini) | LVBench (mini) |
|---|---|---|---|
| | | M-Avg | Overall |
| GPT-4o (OpenAI, 2024a) | ✗ | 59.7 | 31.3 |
| | ✓ | 62.3 (+4.4%) | 32.7 (+4.5%) |
| GPT-5 (OpenAI, 2025) | ✗ | 61.7 | 34.3 |
| | ✓ | 63.3 (+2.6%) | 36.3 (+5.8%) |
| Gemini-2.5-Pro (DeepMind, 2025) | ✗ | 73.3 | 65.7 |
| | ✓ | 76.3 (+4.1%) | 68.7 (+4.6%) |

proves under all metrics on Charades-STA (Gao et al., 2017) under zero-shot setting, suggesting the effectiveness of improving the robustness of the model when facing moments with different lengths.

**Effect of the Verifier for Zoom-in Evaluation** To quantify the verifier's corrective gain, we provide a comparison between w. and w/o the verifier on Charades-STA (Gao et al., 2017) in Table 22. The results demonstrate that the verifier consistently enhances temporal grounding performance, especially on high-quality predictions (*e.g.*, 18.5% higher R@0.7 on the 2B variant), highlighting its importance in the overall pipeline.

**Design Choices of Verifier** In Table 21, we examine various design choices for the verifier. The term "Direct" refers to the method where the grounded moment is directly sent into the model without any expansion. "Expand" denotes expanding the temporal boundaries by 50%, while "Textual" involves adding supplementary textual information to indicate the length of the target event. "Special Token" represents our final approach, utilizing special tokens to denote the grounded start and end timestamps. The comparison demonstrates that expanding the temporal boundaries effectively broadens the verifier's perceptual range, and the use of special tokens enhances the model's awareness of precise moment boundaries.

**Reliability of the Planner** We provide an in-depth investigation into the reliability of the planner. Specifically, we randomly split the planner's training dataset into an 80% training set and a 20% test set, and then re-train the planner on the training set and evaluate it as a three-way classification task on the held-out test set. The metric `planning accuracy` is defined as the proportion of samples for which the predicted reasoning plan is correct. The comparison among different input combinations in Table 23 demonstrate that incorporating both video (even with low resolution) and question input substantially improves planning performance, and the resulting 93% accuracy reflects the considerable reliability of the planner.

**Inference-Time Efficiency** In Table 24, we study the inference-time efficiency of our method on CG-Bench (Chen et al., 2024a). All experiments are conducted on a single NVIDIA RTX 6000 Ada GPU. Compared with the text-based reasoning baseline LongVILA-R1 (Chen et al., 2025b), our full pipeline is approximately 8.9% slower. However, this gap can be easily bridged by activating the planner's auto-planning capability. When the planner is allowed to choose the reasoning path, some easy questions are routed directly to the answerer, which substantially reduces the average inference time from 9.53s to 8.07s per video, resulting 7.8% faster inference speed than the baseline.

**Overall Robustness and Error Accumulation** We acknowledge that the proposed sequential reasoning pipeline has the potential risk of error propagation and accumulation. To quantify this effect, we conduct a systematic analysis of error propagation on two representative datasets: ReXTime (Chen et al., 2024c) (more temporal-related) and NExT-GQA (Xiao et al., 2024) (more reasoning-related). For both datasets, each error case is categorized into one of the following types: (1) **Planning Error:** The question can only be correctly answered by switching to another reasoning plan (*e.g.*, from "all roles" to "answerer only"); (2) **Grounding Error:** All the top-5 predicted moments are incorrect (*i.e.*, having temporal IoU $< 0.5$); (3) **Verification Error:** The moment selected after verification is incorrect; (4) **Answering Error:** The predicted answer is incorrect.

We present the case distributions in Table 19 and error distributions in Figure 7. Several conclusions can be drawn from the results: (1) The planner is highly reliable, accounting for less than 5% of the error cases on both datasets; (2) Grounding errors account for roughly half of all failures. This is aligned with our hypothesis that accurate temporal grounding plays a crucial role in the multi-role reasoning pipeline; (3) Verification and answering contribute comparably smaller portions of the failures, accounting for only about 20% error cases each.

**Correlation between Grounding IoU and QA Accuracy** We study the correlation between temporal grounding performance and QA accuracy in Figure 8. Specifically, we group the samples in ReXTime (Chen et al., 2024c) and NExT-GQA (Xiao et al., 2024) datasets into different IoU buckets, and plot the average QA accuracy within each bucket. On ReXTime, which is more temporal-related, the results show a clear positive correlation between grounding IoU and final QA accuracy. On the more reasoning-related NExT-GQA, such correlation is less significant.

**Controlled Experiments on Closed-source APIs** The results of closed-source models in Table 2 and Table 6 are reported from the corresponding benchmark papers, without strictly aligned settings. Therefore, we provide a controlled experiment to validate the advantages of our method. Specifically, we select two challenging long video understanding benchmarks, *i.e.*, MLVU (Zhou et al., 2024) and LVBench (Wang et al., 2024c), and randomly sample 300 QA pairs from each, forming MLVU (mini) and LVBench (mini). We align the key hyperparameters as follows:

1. **Frame Rate:** 1 FPS
2. **Max Frame Count:** 150
3. **Frame Resolution:** max $448 \times 448$ pixels with natural aspect ratio
4. **Model Hyperparameters:** `temperature = 0, top_p = 0, top_k = 0`

The comparisons are presented in Table 25, clearly showing that our VideoMind-7B outperforms both GPT-4o (OpenAI, 2024a) and Gemini-1.5-Pro (Reid et al., 2024) on both datasets.

**Integration with More Open-source LMMs** In Table 26, we study whether the proposed Chain-of-LoRA pipeline provides a consistent benefit across different base models. The results show that when integrated with stronger base models like Qwen2.5-VL (Bai et al., 2025) and InternVL3 (Zhu et al., 2025), the performance of our Chain-of-LoRA pipeline could be further enhanced on multiple long video benchmarks, highlighting our method's generalizability.

**Integration with Closed-source LMMs** We are also interested in whether the proposed multi-role pipeline could be simulated via a series of prompts on closed-source models. To investigate this, we evaluate the effectiveness of the multi-role reasoning prompt when applied to three models, *i.e.*, GPT-4o (OpenAI, 2024a), GPT-5 (OpenAI, 2025), and Gemini-2.5-Pro (DeepMind, 2025), on the previously constructed MLVU (mini) and LVBench (mini). The results in Table 27 show that our pipeline consistently boosts the performance of different closed-source models, highlighting an interesting finding: the multi-role reasoning pipeline itself can systematically enhance long video reasoning, even without role-specific model designs or training.

## C    MISCELLANEOUS

### C.1    PROMPT TEMPLATES

We present the prompts used in this work, including the input prompts for each role of VideoMind and the prompt for GPT-4o mini (OpenAI, 2024a) for data annotation.

**Prompt for the Planner:**

> You are acting as the planner now. Given a question about the video, your task is to analyze the question and identify the best way to answer this question. You have access to the following tools:
>
> Grounder: Accepts a text query and localizes the relevant video segment according to the query.
> Verifier: A tool supporting grounder by verifying the reliability of its outputs.
> Answerer: Answer a given question directly based on the whole video or a cropped video segment.
>
> Your response must be a list in JSON format. A valid plan for reasoning could be "grounder, verifier, answer", "grounder, verifier", or "answerer", depending on the given question. Please see an example of the format below.
>
> [{"type": "grounder", "value": "text query"}, {"type": "verifier"}, {"type": "answerer"}]
>
> Note that only the grounder can accept an argument called "value", which is the text query used for grounding. Now I give you the question: "**{question}**". Please think carefully and respond with your plan in JSON directly.

**Prompt for the Grounder:**

> You are acting as the grounder now. Given a video and a text query, your goal is to temporally localize the video moment described by the query. If the query is directly describing a moment, simply localize it according to its content. Otherwise, if the moment is described as "before/after a pivotal event", you need to determine the actual event it refers to. The localized moment should only cover the target event. Now I give you the query: "**{query}**". Please think carefully and provide your response.

**Prompt for the Verifier:**

> You are acting as the verifier now. You will be presented a text query describing a moment that potentialy happens in the given video. Your task is to identify whether the video segment between `<SEG-START>` and `<SEG-END>` perfectly covers the moment. If the described moment can be seen in the video, please focus on verifying whether the moment starts at `<SEG-START>` and ends at `<SEG-END>`. Respond with "Yes" if you think the moment boundaries are correct, otherwise "No". If the described moment cannot be seen in the video, respond with "No" directly. Now I give you the query: "**{query}**". Please think carefully and respond with "Yes" or "No" directly.

**Prompt for the Answerer:** When subtitles are considered, we only present the first 100 lines.

> You are given a video with **{duration}** seconds long.
> Subtitles: **{subtitles}**
> **{question}**
> Options:
> (A) **{option 1}**
> (B) **{option 2}**
> (C) **{option 3}**
> (D) **{option 4}**
> Please only give the best option.

**Prompt for Query Rephrasing Data Generation:**

You are an expert in rewriting questions into queries. I will give you a question that requires to be answered based on a specific moment in a video. Your task is to analyze the question and rewrite it into a declarative sentence, which could be used as a text query to search for the relevant video moment. The query should be concise, describing the key event or key scene that the question asks for.

Here are some examples:

Question: How does the male cyclist react when he sees the steep path?
Query: The male cyclist sees the steep path.

Question: What did the girl do at the end of the video?
Query: The end of the video.

Question: What did the lady do as she was cycling off?
Query: The lady is cycling off.

Question: What is the person with red shirt doing on the yacht?
Query: The person with red shirt stays on the yacht.

Now I give you the question: "**{question}**". Please think carefully and respond with the query directly.

## D    THE USE OF LLMS STATEMENT

Large Language Models (LLMs) were used in this study to aid in polishing the manuscript. Specifically, we used LLMs to assist in refining the language and detecting potential grammatical errors. This is to improve readability and ensure clarity of the paper. We confirm that LLMs were not involved in research ideation, method exploration, and experiment designs. All research ideas, methods, and analysis were produced by the authors. We take full responsibility for the content in this paper, including the text generated or polished by the LLMs.

