# OpenReview forum: "VideoMind: A Chain-of-LoRA Agent for Temporal-Grounded Video Reasoning"
_ICLR.cc/2026/Conference — ICLR 2026 Poster_

### Official Review · Reviewer_ocUG · 2025-10-15

**Soundness:** 3
**Presentation:** 3
**Contribution:** 3
**Rating:** 6
**Confidence:** 4

**Summary:**

This paper proposes VideoMind, a video-language agent for temporal-grounded reasoning in long-form videos. The method's core is an agentic workflow comprising four distinct roles—Planner, Grounder, Verifier, and Answerer—that systematically decompose and address complex video queries. To efficiently integrate these roles, the authors introduce a novel "Chain-of-LoRA" mechanism, which leverages a unified base model with lightweight, switchable LoRA adapters for each role. The paper demonstrates the effectiveness of this approach through extensive experiments on benchmarks across three tasks: Grounded VideoQA, Video Temporal Grounding, and General VideoQA.

**Strengths:**

* The "Chain-of-LoRA" mechanism is a key strength, enabling a single model to efficiently switch between specialized roles. This design achieves a strong balance between performance and memory efficiency, making it a more practical alternative to deploying multiple, full models.

* The paper is supported by a thorough and comprehensive evaluation across 14 diverse benchmarks spanning three different video understanding tasks. The claims are well-substantiated by detailed ablation studies that validate the contributions of individual components and the overall framework design.

**Weaknesses:**

* The paper lacks a granular, independent evaluation of each module's reliability. For instance, there is no isolated analysis of the Planner's accuracy in task decomposition, the Grounder's precision, or the Verifier's classification performance. This makes it difficult to pinpoint sources of error or understand the individual limitations of each component.

* The design of the Timestamp Decoder appears overly complex, and the paper fails to provide sufficient justification for this complexity. There is no direct, apple-to-apple comparison against simpler, more common alternatives (e.g., directly predicting timestamps through language modeling), making it difficult to assess the true benefit of the proposed decoder architecture.

* The framework's reliance on an external model (GPT-4o) for query rephrasing undermines its contribution as a self-contained solution. This dependency introduces potential reproducibility issues and reliance on a proprietary API.

* The paper overlooks several recently published and highly relevant papers in video temporal grounding [1-4].

[1] Universal Video Temporal Grounding with Generative Multi-modal Large Language Models

[2] TRACE: Temporal Grounding Video LLM via Causal Event Modeling

[3] Grounded-VideoLLM: Sharpening Fine-grained Temporal Grounding in Video Large Language Models

[4] LLaVA-ST: A Multimodal Large Language Model for Fine-Grained Spatial-Temporal Understanding

**Questions:**

Please see Weaknesses

---

> ### Author Response · Authors · 2025-11-22
> **Response to Reviewer ocUG (1/3)**
>
> Thank you for your recognition of the Chain-of-LoRA design and the comprehensive evaluation! We also sincerely appreciate your detailed and constructive comments. Below we provide point-by-point responses to address your concerns.
>
> ---
>
> **Q1: Provide a granular, independent evaluation of each module's reliability.**
>
> Thanks for the great insight! We provide in-depth studies of the reliabilities of planner, grounder, and verifier as follows.
>
> **Planner:**
>
> We randomly split the planner's training dataset into an 80% training set and a 20% test set, and then re-train the planner on the training set and evaluate it as a **three-way classification** (three types of reasoning plans) task on the held-out test set. The metric `planning accuracy` is defined as the proportion of samples for which the predicted reasoning plan is correct.
>
> We consider three variants of the planner: `video-only`, `question-only`, and `video + question`. The evaluation results are shown below:
>
> | Input Video | Input Question | Planning Acc |
> |:-:|:-:|:-:|
> | ✓ | | 0.42 |
> | | ✓ | 0.79 |
> | ✓ | ✓ | **0.93** |
>
> The results demonstrate that incorporating both video (even with low resolution) and question input substantially improves planning performance, and **the resulting 93% accuracy reflects the considerable reliability of the planner**.
>
> **Grounder:**
>
> Please kindly refer to **Q2** below for a detailed study of the grounder's design. Thank you!
>
> **Verifier:**
>
> We first report the grounder's zero-shot VTG performance on `Charades-STA` dataset:
>
> | Role(s) | Size | R1\@0.3 | R1\@0.5 | R1\@0.7 | R3\@0.3 | R3\@0.5 | R3\@0.7 | R5\@0.3 | R5\@0.5 | R5\@0.7 | mIoU |
> |:-:|:-:|:-:|:-:|:-:|:-:|:-:|:-:|:-:|:-:|:-:|:-:|
> | Grounder | 2B | 63.2 | 46.9 | 20.5 | 77.7 | 66.3 | 42.2 | 84.4 | 76.5 | 53.1 | 41.7 |
> | Grounder | 7B | 69.4 | 53.2 | 26.6 | 81.4 | 71.2 | 47.7 | 86.6 | 79.8 | 57.9 | 46.8 |
>
> From the results, we observe that sometimes the grounder's top-1 predictions may be unreliable, while **the top-5 predictions yield much higher recall** (*e.g.,* `R1@0.5=46.9` *vs.* `R5@0.5=76.5` for the 2B variant). Motivated by this, we introduce a verifier to re-rank the top-5 results by zooming in for fine-grained investigation. The comparison between w. and w/o the verifier is as follows.
>
> | Role(s) | Size | &nbsp;&nbsp;&nbsp;&nbsp;&nbsp;R1\@0.3&nbsp;&nbsp;&nbsp;&nbsp;&nbsp; | &nbsp;&nbsp;&nbsp;&nbsp;&nbsp;R1\@0.5&nbsp;&nbsp;&nbsp;&nbsp;&nbsp; | &nbsp;&nbsp;&nbsp;&nbsp;&nbsp;R1\@0.7&nbsp;&nbsp;&nbsp;&nbsp;&nbsp; | &nbsp;&nbsp;&nbsp;&nbsp;&nbsp;mIoU&nbsp;&nbsp;&nbsp;&nbsp;&nbsp; | IoU Raise |
> |-|:-:|:-:|:-:|:-:|:-:|:-:|
> | Grounder | 2B | 63.2 | 46.9 | 20.5 | 41.7 | -- |
> | Grounder + Verifier | 2B | **68.0** (+7.6%) | **51.2** (+9.2%) | **24.3** (+18.5%) | **44.8** (+7.4%) | **32.9%** |
> | Grounder | 7B | 69.4 | 53.2 | 26.6 | 46.8 | -- |
> | Grounder + Verifier | 7B | **73.8** (+6.3%) | **59.1** (+11.1%) | **30.1** (+13.2%) | **49.8** (+6.4%) | **31.3%** |
>
> Here, `IoU Raise` means the percentage of the samples whose grounding IoU is raised by the verifier. The results demonstrate that **the verifier consistently enhances temporal grounding performance**, especially on **high-quality predictions** (*e.g.,* 18.5% higher `R1@0.7` for the 2B variant), highlighting its importance in the overall system.
>
> The experiments and discussions above have been added to `Sec. B.3`, `Table 23`, `Table 18`, and `Table 22`.

---

> ### Author Response · Authors · 2025-11-22
> **Response to Reviewer ocUG (2/3)**
>
> **Q2: Provide sufficient justification for the complexity of the timestamp decoder.**
>
> Thanks for pointing this out! We acknowledge that introducing a timestamp decoder inevitably brings extra complexity. However, this is justified as the grounder plays a crucial role in our proposed Chain-of-LoRA pipeline. **The model's temporal grounding quality directly impacts the final QA accuracy.** To demonstrate the necessity of this design, we implement and compare the following alternative timestamp modeling techniques based on VideoMind-2B (Grounder):
>
> 1. **Text-only** [5]: Directly represent timestamps in text form (*e.g.,* "2.3 seconds").
> 2. **Special Tokens** [6]: Define a set of timestamp tokens (*e.g.,* `<T0>`, `<T1>`, ... `<TN>`).
> 3. **Embedding Matching** [7]: Predict frame-level features and retrieve the closest frame index via similarity.
> 4. **Time Marker** [8]: Explicitly insert textual timestamps among visual tokens.
>
> Zero-shot VTG performance on `Charades-STA` dataset:
>
> | Timestamp Modeling Method | R\@0.3 | R\@0.5 | R\@0.7 | mIoU |
> |-|:-:|:-:|:-:|:-:|
> | Text-only [5] | 56.8 | 39.5 | 14.3 | 36.1 |
> | Special Tokens [6] | 56.4 | 39.2 | 14.5 | 35.7 |
> | Embedding Matching [7] | 59.6 | 43.5 | 17.0 | 38.2 |
> | Time Marker [8] | 60.5 | 43.9 | 17.2 | 38.6 |
> | Timestamp Decoder (Ours) | **64.1** | **47.2** | **21.7** | **42.0** |
>
> The results clearly demonstrate that **the timestamp decoder delivers the strongest temporal grounding capability**. We attribute it to two key factors:
>
> 1. It decouples continuous timestamp modeling from discrete token prediction, allowing the model to represent time with higher precision.
> 2. The direct regression supervision (L1 Loss) further enhances time reasoning and stabilizes training.
>
> Moreover, the timestamp decoder naturally supports predicting **multiple moments with corresponding confidence scores**, supporting tasks like multi-moment retrieval (QVHighlights [9]) and facilitating moment re-ranking through the verifier. These advantages jointly enhance the reliability of temporal grounding, which ensures the correct moment can be localized for further reasoning.
>
> The experiment and discussion above have been added to `Sec. B.3` and `Table 18`.
>
> ---
>
> **Q3: The framework's reliance on GPT-4o for query rephrasing undermines its contribution as a self-contained solution.**
>
> We kindly clarify that GPT-4o mini is only used for **training data generation** during the planner's development phase. It is not used during inference. Therefore, our solution operates **fully self-contained**, without dependence on any external LLMs/APIs at test time. We have revised the relevant statements in `Sec. 3.1` to ensure no misleading.
>
> In greater detail, during data generation, GPT-4o mini is utilized to rewrite **complex questions** (*e.g.,* "What is the person sitting on the bed doing as the baby plays?") into **simple moment queries** (*e.g.,* "the baby is playing"), which are easier for the grounder to parse. We train the planner on such question-query pairs to enable its query-rephrasing capability.

---

> ### Author Response · Authors · 2025-11-22
> **Response to Reviewer ocUG (3/3)**
>
> **Q4: The paper overlooks several recently published and highly relevant papers in video temporal grounding [1-4].**
>
> Thanks for highlighting these important baselines! We acknowledge that they are all highly competitive MLLMs focusing on video temporal grounding. Their results on `Charades-STA` and `ActivityNet-Captions` datasets are compared as follows.
>
> Performance on `Charades-STA` dataset:
>
> | Method | Size | Training Data | Zero-shot | R\@0.3 | R\@0.5 | R\@0.7 | mIoU |
> |-|:-:|:-:|:-:|:-:|:-:|:-:|:-:|
> | TRACE [2] | 7B | 635K | ✓ | -- | 40.3 | 19.4 | -- |
> | Grounded-VideoLLM [3] | 4B* | 920K | ✓ | 54.2 | 36.4 | 19.7 | 36.8 |
> | LLaVA-ST [4] | 7B | 920K | ✓ | 63.1 | 44.8 | 23.4 | 42.4 |
> | UniTime [1] | 7B | 1.3M | ✓ | -- | **59.1** | **31.9** | **52.2** |
> | VideoMind (Ours) | 7B | 210K | ✓ | **73.5** | **59.1** | 31.2 | 50.2 |
>
> Performance on `ActivityNet-Captions` dataset:
>
> | Method | Size | Training Data | Zero-shot | R\@0.3 | R\@0.5 | R\@0.7 | mIoU |
> |-|:-:|:-:|:-:|:-:|:-:|:-:|:-:|
> | TRACE [2] | 7B | 635K | | -- | 37.7 | 24.0 | 39.0 |
> | Grounded-VideoLLM [3] | 4B* | 920K | | 46.2 | 30.3 | 19.0 | 36.1 |
> | UniTime [1] | 7B | 1.3M | ✓ | -- | 22.8 | 14.1 | 27.3 |
> | VideoMind (Ours) | 7B | 210K | ✓ | **48.4** | **30.3** | **15.7** | **33.3** |
>
> Note that:
>
> - `Training Data` indicates the number of timestamp-related samples used during training.
> - `Zero-shot` means neither the evaluation annotations nor videos were seen during training.
> - For Grounded-VideoLLM, we report results from its 4B (Phi-3.5-V) rather than the 7B (LLaVA-1.5) variant, as it performs better on these datasets.
>
> On `Charades-STA`, VideoMind-7B outperforms all the other models except UniTime-7B. The small performance gap (-2.8% mIoU) is reasonable, given that UniTime-7B was trained on a dataset **over 6x larger** than ours.
>
> On `ActivityNet-Captions`, only the UniTime is fairly comparable, as both TRACE and Grounded-VideoLLM incorporate videos/annotations from `ActivityNet-Captions` during training. VideoMind-7B performs **significantly better than UniTime-7B** (+22.0% mIoU) on this dataset.
>
> These methods and their experimental results have been cited and added to `Table 5`.
>
> ---
>
> **References:**
>
> [1] Universal Video Temporal Grounding with Generative Multi-modal Large Language Models, NeurIPS 2025
>
> [2] TRACE: Temporal Grounding Video LLM via Causal Event Modeling, ICLR 2025
>
> [3] Grounded-VideoLLM: Sharpening Fine-grained Temporal Grounding in Video Large Language Models, EMNLP 2025 Findings
>
> [4] LLaVA-ST: A Multimodal Large Language Model for Fine-Grained Spatial-Temporal Understanding, CVPR 2025
>
> [5] TimeChat: A Time-sensitive Multimodal Large Language Model for Long Video Understanding, CVPR 2024
>
> [6] Momentor: Advancing Video Large Language Model with Fine-Grained Temporal Reasoning, ICML 2024
>
> [7] E.T. Bench: Towards Open-Ended Event-Level Video-Language Understanding, NeurIPS 2024
>
> [8] TimeMarker: A Versatile Video-LLM for Long and Short Video Understanding with Superior Temporal Localization Ability, arXiv 2024
>
> [9] QVHighlights: Detecting Moments and Highlights in Videos via Natural Language Queries, NeurIPS 2021

---

> > ### Comment · Reviewer_ocUG · 2025-11-25
> >
> > Thanks for the authors' explanations. I have no questions, and I keep my positive assessment for this paper.

---

> > > ### Author Response · Authors · 2025-11-25
> > >
> > > Dear Reviewer ocUG,
> > >
> > > Thank you very much for your feedback!
> > >
> > > We are glad to hear that your concerns have been addressed. Thanks again for reviewing our paper!
> > >
> > > Sincerely,
> > >
> > > Authors of 1067

---

### Official Review · Reviewer_3Ldi · 2025-10-16

**Soundness:** 3
**Presentation:** 3
**Contribution:** 3
**Rating:** 6
**Confidence:** 3

**Summary:**

The paper proposes VideoMind, a unified multimodal agent framework for long-video temporal grounding and question answering. It decomposes capabilities into four roles—Planner, Grounder, Verifier, and Answerer—and employs a Chain-of-LoRA strategy to efficiently invoke different roles by switching LoRA adapters on the same base large language model. The model is evaluated on 14 datasets spanning Grounded VideoQA, Video Temporal Grounding, and general VideoQA, reporting state-of-the-art or competitive results on multiple long-video benchmarks. The authors also commit to releasing the code and models.

**Strengths:**

1.	The method is structurally clear and engineering-efficient: it modularizes planning, temporal localization, verification, and answering, and realizes one model with multiple roles via LoRA switching, which reduces GPU memory while keeping flexible composition—an appealing design.
2.	The Grounder includes a decoder specialized for temporal boundaries: a <REG>-triggered timestamp decoder and a multi-scale temporal pyramid, avoiding reliance on language tokens alone for time prediction; the technique is solid.
3.	The Verifier adopts zoom-out recheck plus boundary markers: it expands both sides of a candidate segment and inserts <SEG-START/END>, performing binary validation to improve IoU and robustness, with clear training and inference definitions.
4.	Efficiency–effectiveness trade-off: compared with multi-model dispatch, Chain-of-LoRA achieves comparable accuracy while significantly reducing memory usage (4.2 GB vs. 16.6 GB), validating the design intent.

**Weaknesses:**

1.	Novelty boundary: While the multi-role agent and LoRA switching are practical, their difference from existing modular tool-based agents with vision APIs/components lies mainly in implementation form rather than a new paradigm; stronger theoretical grounding or unified learning evidence is needed to highlight originality.
2.	 Joint training and error propagation: The authors acknowledge in limitations/future work that better joint optimization is required. The current pipeline-style chaining of roles may accumulate upstream errors, and the paper lacks quantitative analysis of error propagation.
3.	Reproducibility details and open-source dependencies: Although the appendix lists training hyperparameters and dataset splits, implementation details for data reuse/annotation across roles remain brief. It is recommended to provide role-level prompts, sampling, and filtering scripts in the supplementary material to ensure others can reproduce the key results in Tables 2/3/5.
4.	Choice of baselines and strength of claims: Comparisons to GPT-4o and Gemini-1.5-Pro emphasize advantages on portions of long-video benchmarks, but input/context configurations and sampling strategies for these closed models are not fully aligned. Controlled comparisons with equal frame sampling, total frame count, and resolution are suggested.
5.	Cross-domain generalization and failure cases: On domain-specific datasets such as Ego4D-NLQ, performance lags behind strong pre-trained specialists; the paper provides only brief explanations and lacks targeted analyses and error visualizations.

**Questions:**

1.	Have you attempted multi-task training that simultaneously optimizes the Planner/Grounder/Verifier on a shared backbone? If such attempts were unsuccessful, please share the main challenges and stabilization techniques.
2.	Error sensitivity: Could you provide a sensitivity curve from the Grounder’s mIoU to the final QA accuracy, or bucket results on NExT-GQA by candidate-segment IoU to quantify the Verifier’s corrective gain?
3.	Resource–performance trade-off: While Chain-of-LoRA has low VRAM usage, what is the time cost of repeated “zoom-out rechecks” and multi-candidate verification during inference? Can you report wall-clock comparisons and throughput curves on CG-Bench?
4.	 Fair comparison with closed-source LMMs: Are frame sampling for long videos, maximum frame count, subtitle usage, temperature/sampling steps, etc., aligned? Could you include strictly controlled settings in the appendix to substantiate the stronger claim that the 2B/7B variants surpass GPT-4o/Gemini?

---

> ### Author Response · Authors · 2025-11-22
> **Response to Reviewer 3Ldi (1/4)**
>
> Many thanks for your careful review and insightful comments! We are encouraged by your recognition of our structurally clear design, solid technique, and efficiency–effectiveness trade-off. Below we provide responses to your concerns in detail.
>
> ---
>
> **Q1: Highlight the originality of VideoMind compared with existing modular tool-based agents with vision APIs/components.**
>
> Thanks for your comment! We acknowledge that our proposed multi-role agent shares the high-level idea of decomposing complex problems into step-by-step solvable sub-tasks. Nevertheless, **VideoMind introduces several non-trivial innovations** that distinguish it from conventional tool-based systems with vision APIs/components:
>
> 1. We design **a unified LMM backbone with role-specific LoRA adapters**. Unlike existing agent systems that rely on **heterogeneous** external APIs/modules (*e.g.,* detectors, captioners, or RAG modules), VideoMind implements all the four roles based on a unified LMM backbone.
> 2. Our proposed framework supports **flexible inference-time role switching with minimal memory overhead**, which is more efficient and flexible compared to deploying multiple LLMs/LMMs or calling different APIs.
> 3. The **grounder-verifier combination itself is a state-of-the-art video temporal grounding solution**. This includes several novel technical designs including *a dedicated timestamp decoder*, *a temporal feature pyramid*, and *a zoom-in verification strategy*, enabling robust zero-shot temporal grounding.
>
> We have revised the relevant statements in `Sec. 1` and `Sec. 2`. More detailed experiments and discussions regarding the significance of our pipeline have also been provided in `Sec. B.3`.
>
> ---
>
> **Q2: Provide quantitative analysis of the error propagation and accumulation within the pipeline.**
>
> Thanks for the valuable insight! We conduct a systematic quantitative analysis of error propagation on two representative datasets: `ReXTime` (more temporal-related) and `NExT-GQA` (more reasoning-related). For both datasets, each error case is categorized into one of the following types:
>
> 1. **Planning Error:** The question can only be correctly answered by switching to another reasoning plan (*e.g.,* from "all roles" to "answerer only").
> 2. **Grounding Error:** Not in `1`, but all the top-5 predicted moments are incorrect (*i.e.,* temporal IoU < 0.5).
> 3. **Verification Error:** Not in `1` + `2`, but the moment selected after verification is incorrect.
> 4. **Answering Error:** Not in `1` + `2` + `3`, but the predicted answer is incorrect.
>
> Case distribution on `ReXTime` dataset:
>
> | Method | Size | Correct | Planning Error | Grounding Error | Verification Error | Answering Error |
> |:-:|:-:|:-:|:-:|:-:|:-:|:-:|
> | VideoMind | 2B | 69.1% | 1.2% | 18.3% | 5.7% | 5.7% |
> | VideoMind | 7B | 74.6% | 1.1% | 15.0% | 4.6% | 4.7% |
>
> Case distribution on `NExT-GQA` dataset:
>
> | Method | Size | Correct | Planning Error | Grounding Error | Verification Error | Answering Error |
> |:-:|:-:|:-:|:-:|:-:|:-:|:-:|
> | VideoMind | 2B | 71.2% | 1.9% | 14.0% | 6.9% | 6.0% |
> | VideoMind | 7B | 76.6% | 0.7% | 11.8% | 5.8% | 5.1% |
>
> Several conclusions can be drawn from the distributions:
>
> 1. **The planner is highly reliable**, with less than 2% error rate on both datasets.
> 2. **Grounding errors account for roughly half of all failures.** This is aligned with our hypothesis that accurate temporal grounding plays a crucial role in our proposed pipeline.
> 3. **Verification and answering contribute comparably smaller portions of the failures**, accounting for only about 5% error rate each.
>
> Overall, these results highlight that the temporal grounding stage (grounder + verifier) dominates the pipeline's error accumulation. Strengthening grounding capabilities and introducing appropriate reflection mechanisms are potentially the most effective paths toward improving end-to-end long video reasoning.
>
> The experiments and discussion above have been added to `Sec. B.3`, `Table 19`, and `Figure 7` **(Error Distribution Visualization)**.

---

> ### Author Response · Authors · 2025-11-22
> **Response to Reviewer 3Ldi (2/4)**
>
> **Q3: It is recommended to provide more implementation details about data reuse/annotation across roles.**
>
> Thanks for raising this important point! We are committed to making our research fully open-source and reproducible. To achieve this, we release the following materials:
>
> 1. **Source Code:** This includes the code for the models, training, evaluation, and data pre-processing.
> 2. **Checkpoints & Training Logs:** For `VideoMind-2B` and `VideoMind-7B`.
> 3. **Datasets:** Including raw videos, compressed videos, annotations, and pre-processing scripts for 27 video grounding/QA datasets in this project.
> 4. **Online Demo:** There will be an online interactive demo to showcase the key capabilities of VideoMind.
> 5. **Others:** We also release the data used during our early exploration (but not included in the final version) to facilitate future research.
>
> The source code and a demo video were already included in the initial submission.
>
> > It is recommended to provide role-level prompts, sampling, and filtering scripts in the supplementary material to ensure others can reproduce the key results in Tables 2/3/5.
>
> We appreciate this suggestion. The role-level prompts have been provided in `Sec. C.1`. The scripts for planner's training data generation (`api_rewrite_gpt.py` and `gen_planning_data.py`) have been uploaded to the supplementary material. We confirm that **all the results in** `Tables 2/3/5` **can be reproduced using our open-source code and data**.
>
> ---
>
> **Q4: Controlled comparisons to GPT-4o & Gemini-1.5-Pro with equal frame sampling, total frame count, and resolution are suggested.**
>
> Great suggestion! To provide such a comparison, we select two challenging long video understanding benchmarks, *i.e.,* `MLVU` and `LVBench`, and randomly sample 300 QA pairs from each. The key hyperparameters are fixed as follows.
>
> 1. **Frame Sampling Method:** 1 FPS
> 2. **Max Frame Count:** 150
> 3. **Frame Resolution:** max 448 $\times$ 448 pixels with natural aspect ratio
> 4. **Model Hyperparameters:** `temperature = 0`, `top_p = 0`, `top_k = 0`, `max_new_tokens = 256`
>
> We present the evaluation results below:
>
> | Method | Size | MLVU-mini$_{Acc}$ | LVBench-mini$_{Acc}$ |
> |-|:-:|:-:|:-:|
> | GPT-4o | -- | 59.7 | 31.3 |
> | Gemini-1.5-Pro | -- | 60.3 | 36.3 |
> | VideoMind (Ours) | 2B | 59.3 | 35.7 |
> | VideoMind (Ours) | 7B | **62.7** | **40.3** |
>
> The results from strictly controlled experiments verify that **VideoMind-7B outperforms GPT-4o and Gemini-1.5-Pro on both long video benchmarks**, demonstrating the significance and effectiveness of the proposed vision-centric reasoning pipeline.
>
> The experiment and discussion above have been added to `Sec. B.3` and `Table 25`.
>
> ---
>
> **Q5: VideoMind's performance lags behind strong pre-trained specialists on domain-specific datasets such as Ego4D-NLQ.**
>
> Thanks for your detailed observation. We clarify that in `Table 12` (Results on Ego4D-NLQ), **UniVTG [1] is the only fairly comparable baseline**, as all other methods were **directly trained** on Ego4D-NLQ. We acknowledge that only VideoMind-7B outperforms UniVTG but VideoMind-2B does not. This is justifiable as **neither the grounder nor the verifier was trained on egocentric videos**, while UniVTG was pretrained on **1.8M samples from Ego4D [2]**.
>
> To align the setting, we trained an additional VideoMind-2B variant with **extra 67K grounding samples from NaQ [3]**. The resulting zero-shot performance on `Ego4D-NLQ` is presented below.
>
> | Method | Size | Egocentric Data | R\@0.3 | R\@0.5 | R\@0.7 | mIoU |
> |-|:-:|:-:|:-:|:-:|:-:|:-:|
> | UniVTG [2] | -- | 1.8M | 6.5 | 3.5 | 1.2 | 4.6 |
> | VideoMind (Ours) | 2B | -- | 5.9 | 2.9 | 1.2 | 4.7 |
> | VideoMind (Ours) | 2B | 67K | **7.2** | **3.9** | **1.8** | **5.3** |
>
> The results show that the temporal grounding performance can be effectively enhanced through co-training the model on a small amount of egocentric data.
>
> The experiment and discussion above have been added to `Sec. B.2` and `Table 12`.

---

> ### Author Response · Authors · 2025-11-22
> **Response to Reviewer 3Ldi (3/4)**
>
> **Q6: Have you attempted multi-task training that simultaneously optimizes the planner/grounder/verifier on a shared backbone?**
>
> Yes. We have reported the performance of `All-in-One` setting in `Table 7`. The results are presented below for your reference.
>
> | &nbsp;&nbsp;&nbsp;&nbsp;&nbsp;Method&nbsp;&nbsp;&nbsp;&nbsp;&nbsp; | NExT-GQA$_{mIoU}$ | NExT-GQA$_{Acc}$ | Charades-STA$_{R\@0.5}$ | Charades-STA$_{mIoU}$ | Video-MME$_{All}$ | Video-MME$_{Long}$ |
> |:-:|:-:|:-:|:-:|:-:|:-:|:-:|
> | All-in-One | 28.0 | 70.5 | 47.8 | 42.1 | 53.6 | 43.6 |
> | Chain-of-LoRA | **28.6** | **71.4** | **51.1** | **45.2** | **55.4** | **46.3** |
>
> The comparison clearly demonstrates that **decoupling the roles into separate LoRA adapters yields significantly better performance** than jointly optimizing all roles on a shared backbone. This is because the key capabilities of different roles may sometimes **conflict with one another**, and our Chain-of-LoRA mechanism offers **the best balance between effectiveness and efficiency**.
>
> The discussion above has been added to `Sec. 4.2`.
>
> ---
>
> **Q7: Could you provide a sensitivity curve from the grounder's mIoU to the final QA accuracy, or bucket results on NExT-GQA by candidate-segment IoU to quantify the verifier's corrective gain?**
>
> Nice suggestion! We provide the analysis on both `ReXTime` and `NExT-GQA` datasets. The relations between VideoMind-2B's grounding and QA accuracies are shown below.
>
> QA accuracies under different grounding IoUs on `ReXTime` dataset:
>
> | Grounding IoU | &nbsp;0 $\sim$ 0.1&nbsp; | &nbsp;0.1 $\sim$ 0.2&nbsp; | &nbsp;0.2 $\sim$ 0.3&nbsp; | &nbsp;0.3 $\sim$ 0.4&nbsp; | &nbsp;0.4 $\sim$ 0.5&nbsp; | &nbsp;0.5 $\sim$ 0.6&nbsp; | &nbsp;0.6 $\sim$ 0.7&nbsp; | &nbsp;0.7 $\sim$ 0.8&nbsp; | &nbsp;0.8 $\sim$ 0.9&nbsp; | &nbsp;0.9 $\sim$ 1&nbsp; |
> |:-:|:-:|:-:|:-:|:-:|:-:|:-:|:-:|:-:|:-:|:-:|
> | **QA Accuracy** | 0.52 | 0.63 | 0.66 | 0.70 | 0.77 | 0.72 | 0.68 | 0.78 | 0.82 | 0.87 |
>
> QA accuracies under different grounding IoUs on `NExT-GQA` dataset:
>
> | Grounding IoU | &nbsp;0 $\sim$ 0.1&nbsp; | &nbsp;0.1 $\sim$ 0.2&nbsp; | &nbsp;0.2 $\sim$ 0.3&nbsp; | &nbsp;0.3 $\sim$ 0.4&nbsp; | &nbsp;0.4 $\sim$ 0.5&nbsp; | &nbsp;0.5 $\sim$ 0.6&nbsp; | &nbsp;0.6 $\sim$ 0.7&nbsp; | &nbsp;0.7 $\sim$ 0.8&nbsp; | &nbsp;0.8 $\sim$ 0.9&nbsp; | &nbsp;0.9 $\sim$ 1&nbsp; |
> |:-:|:-:|:-:|:-:|:-:|:-:|:-:|:-:|:-:|:-:|:-:|
> | **QA Accuracy** | 0.65 | 0.71 | 0.75 | 0.70 | 0.72 | 0.76 | 0.81 | 0.77 | 0.69 | 0.82 |
>
> On `ReXTime`, which is more temporal-related, the results show a **clear positive correlation between grounding IoU and final QA accuracy**. Higher grounding IoUs consistently lead to better QA performance. This highlights the importance of accurate temporal grounding, as stronger localization directly improves the overall video QA accuracy. On the more reasoning-related `NExT-GQA`, such correlation is less significant.
>
> To quantify the verifier's corrective gain, we provide a comparison between w. and w/o the verifier on `Charades-STA` as follows.
>
> | Role(s) | Size | &nbsp;&nbsp;&nbsp;&nbsp;&nbsp;R1\@0.3&nbsp;&nbsp;&nbsp;&nbsp;&nbsp; | &nbsp;&nbsp;&nbsp;&nbsp;&nbsp;R1\@0.5&nbsp;&nbsp;&nbsp;&nbsp;&nbsp; | &nbsp;&nbsp;&nbsp;&nbsp;&nbsp;R1\@0.7&nbsp;&nbsp;&nbsp;&nbsp;&nbsp; | &nbsp;&nbsp;&nbsp;&nbsp;&nbsp;mIoU&nbsp;&nbsp;&nbsp;&nbsp;&nbsp; | IoU Raise |
> |-|:-:|:-:|:-:|:-:|:-:|:-:|
> | Grounder | 2B | 63.2 | 46.9 | 20.5 | 41.7 | -- |
> | Grounder + Verifier | 2B | **68.0** (+7.6%) | **51.2** (+9.2%) | **24.3** (+18.5%) | **44.8** (+7.4%) | **32.9%** |
> | Grounder | 7B | 69.4 | 53.2 | 26.6 | 46.8 | -- |
> | Grounder + Verifier | 7B | **73.8** (+6.3%) | **59.1** (+11.1%) | **30.1** (+13.2%) | **49.8** (+6.4%) | **31.3%** |
>
> Here, `IoU Raise` means the percentage of the samples whose grounding IoU is raised by the verifier. The results demonstrate that **the verifier consistently enhances temporal grounding performance**, especially on **high-quality predictions** (*e.g.,* 18.5% higher `R1@0.7` on the 2B variant), highlighting its importance in the overall system.
>
> The experiments and discussions above have been added to `Sec. B.3`, `Figure 8` **(Correlation Visualization)**, and `Table 22`.

---

> ### Author Response · Authors · 2025-11-22
> **Response to Reviewer 3Ldi (4/4)**
>
> **Q8: Can you report wall-clock comparisons and throughput curves on CG-Bench?**
>
> Sure and thanks for your suggestion! Below we compare the **average inference time per video on CG-Bench** with a text-based reasoning baseline `LongVILA-R1` [4] on a single NVIDIA RTX 6000 Ada GPU.
>
> | Method | Size | Inference Time (s/video) |
> |-|:-:|:-:|
> | LongVILA-R1 [4] | 7B | 8.75 |
> | VideoMind | 7B | 9.53 (+8.9%) |
> | VideoMind (w. Auto Planning) | 7B | 8.07 (-7.8%) |
>
> Compared with `LongVILA-R1`, our full pipeline is approximately 8.9% slower. However, this gap can be easily bridged by **activating the planner's auto-planning capability**. When the planner is allowed to choose the reasoning path, some easy questions are routed directly to the answerer, which substantially reduces the average inference time from 9.53s to 8.07s per video, resulting **7.8% faster inference speed than the baseline**.
>
> The results and discussion above have been added to `Sec. B.3` and `Table 24`.
>
> ---
>
> **Q9: Provide fair comparisons with closed-source LMMs.**
>
> Please kindly refer to our response to **Q4** above for details. Thank you!
>
> ---
>
> **References:**
>
> [1] UniVTG: Towards Unified Video-Language Temporal Grounding, ICCV 2023
>
> [2] Ego4D: Around the World in 3,000 Hours of Egocentric Video, CVPR 2022
>
> [3] NaQ: Leveraging Narrations as Queries to Supervise Episodic Memory, CVPR 2023
>
> [4] Scaling RL to Long Videos, NeurIPS 2025

---

> ### Author Response · Authors · 2025-11-26
>
> Dear Reviewer 3Ldi,
>
> Many thanks again for your detailed review and constructive comments!
>
> We would greatly appreciate it if you could kindly review our responses to ensure they adequately address your concerns. We remain fully committed to clarifying any remaining points and welcome any further discussions :)
>
> Thank you for your time and consideration.
>
> Sincerely,
>
> Authors of 1067

---

### Official Review · Reviewer_RtrU · 2025-10-26

**Soundness:** 3
**Presentation:** 3
**Contribution:** 3
**Rating:** 8
**Confidence:** 4

**Summary:**

This paper presents VideoMind, a novel video-language agent designed for temporal-grounded video reasoning. The authors identify a key limitation in existing models: while strong at text-based reasoning, they struggle to explicitly localize, revisit, and reason about specific temporal moments in long videos.

**Strengths:**

1. This paper proposes the "Chain-of-LoRA", which is a highly practical solution to a major challenge in agentic AI. It directly addresses the need for multi-skilled agents without the prohibitive memory overhead of loading multiple full models.
2. The paper's claims are backed by a comprehensive set of experiments across 14 benchmarks. The results are consistently strong.

**Weaknesses:**

1. While the 'Chain-of-LoRA' mechanism is shown to be highly memory-efficient, how does the total wall-clock inference time of the full, multi-step VideoMind pipeline (Planner $\rightarrow$ Grounder $\rightarrow$ Verifier $\rightarrow$ Answerer) compare to an end-to-end baseline model that performs reasoning in a single forward pass? Does the paper provide any latency benchmarks (e.g., in seconds per video)?
2. How dependent are the results on the Qwen2-VL architecture? Could the 'Chain-of-LoRA' training approach be applied to other strong open-source VideoLLMs (e.g., InternVL3 or Qwen2.5-VL) to validate that this agentic pipeline provides a consistent benefit across different base models?
3. The workflow itself, separate from the LoRA implementation, could be a significant contribution. Could this multi-step reasoning pipeline (plan $\rightarrow$ ground $\rightarrow$ verify $\rightarrow$ answer) be simulated via a series of prompts on closed-source models like GPT-4o, Gemini 1.5 Pro or newer models? This would help isolate the performance gain of the agentic workflow from the specific model training.

**Questions:**

See Weaknesses.

---

> ### Author Response · Authors · 2025-11-22
> **Response to Reviewer RtrU**
>
> Thank you very much for recognizing the value of our work and for providing the constructive feedback! Below we provide detailed responses to your concerns and suggestions.
>
> ---
>
> **Q1: Compare the total wall-clock inference time of VideoMind and end-to-end reasoning models.**
>
> Thanks for your suggestion! Below we compare the **average inference time per video on CG-Bench** with a text-based reasoning baseline `LongVILA-R1` [4] on a single NVIDIA RTX 6000 Ada GPU.
>
> | Method | Size | Inference Time (s/video) |
> |-|:-:|:-:|
> | LongVILA-R1 [4] | 7B | 8.75 |
> | VideoMind | 7B | 9.53 (+8.9%) |
> | VideoMind (w. Auto Planning) | 7B | 8.07 (-7.8%) |
>
> Compared with `LongVILA-R1`, our full pipeline is approximately 8.9% slower. However, this gap can be easily bridged by **activating the planner's auto-planning capability**. When the planner is allowed to choose the reasoning path, some easy questions are routed directly to the answerer, which substantially reduces the average inference time from 9.53s to 8.07s per video, resulting **7.8% faster inference speed than the baseline**.
>
> The results and discussion above have been added to `Sec. B.3` and `Table 24`.
>
> ---
>
> **Q2: Could the Chain-of-LoRA mechanism be applied to other strong open-source VideoLLMs (*e.g.,* InternVL3 [2] or Qwen2.5-VL [3])?**
>
> Yes. We have tried implementing Chain-of-LoRA on both architectures. The evaluation results are shown below:
>
> | Base Model | Size | CG-Bench$_{acc.\@IoU}$ | ReXTime$_{Acc\@IoU}$ | Video-MME$_{w/o sub.}$ | MLVU$_{Acc}$ | LVBench$_{Acc}$ |
> |-|:-:|:-:|:-:|:-:|:-:|:-:|
> | Qwen2-VL [1] | 2B | 4.0 | 17.3 | 55.4 | 58.7 | 35.4 |
> | Qwen2-VL [1] | 7B | 4.7 | 20.2 | 58.2 | 64.4 | 40.8 |
> | InternVL3 [2] | 2B | 4.1 | 17.5 | 58.2 | 61.4 | 38.1 |
> | InternVL3 [2] | 8B | 4.5 | **20.8** | **66.5** | 63.8 | 42.3 |
> | Qwen2.5-VL [3] | 3B | 5.0 | 15.6 | 60.9 | 62.7 | 40.5 |
> | Qwen2.5-VL [3] | 7B | **5.7** | 19.8 | 65.9 | **66.3** | **45.2** |
>
> With stronger base models like `InternVL3` and `Qwen2.5-VL`, **the performance of our Chain-of-LoRA pipeline could be further enhanced** on multiple long video benchmarks.
>
> The results and discussion above have been added to `Sec. B.3` and `Table 26`.
>
> ---
>
> **Q3: Could the multi-step reasoning pipeline be simulated via a series of prompts on closed-source models?**
>
> Yes and thanks for the interesting suggestion! To investigate it, we select two challenging long video understanding benchmarks, *i.e.,* `MLVU` and `LVBench`, and randomly sample 300 QA pairs from each. We then evaluate the effectiveness of the multi-role reasoning prompt when applied to three closed-source models: `GPT-4o`, `GPT-5`, and `Gemini-2.5-Pro`.
>
> | Method | Multi-Role Pipeline | MLVU-mini$_{Acc}$ | LVBench-mini$_{Acc}$ |
> |-|:-:|:-:|:-:|
> | GPT-4o | | 59.7 | 31.3 |
> | GPT-4o | ✓ | **62.3** (+4.4%) | **32.7** (+4.5%) |
> | GPT-5 | | 61.7 | 34.3 |
> | GPT-5 | ✓ | **63.3** (+2.6%) | **36.3** (+5.8%) |
> | Gemini-2.5-Pro | | 73.3 | 65.7 |
> | Gemini-2.5-Pro | ✓ | **76.3** (+4.1%) | **68.7** (+4.6%) |
>
> The comparison clearly shows that the "planner $\to$ grounder $\to$ verifier $\to$ answerer" pipeline **consistently boosts the performance of different closed-source models**. This highlights an interesting finding: the multi-step reasoning pipeline itself can systematically enhance long/complex video reasoning, **even without role-specific model designs or training**.
>
> The results and discussion above have been added to `Sec. B.3` and `Table 27`.
>
> ---
>
> **References:**
>
> [1] Qwen2-VL: Enhancing Vision-Language Model's Perception of the World at Any Resolution, arXiv 2024
>
> [2] InternVL3: Exploring Advanced Training and Test-Time Recipes for Open-Source Multimodal Models, arXiv 2025
>
> [3] Qwen2.5-VL Technical Report, arXiv 2025

---

> ### Author Response · Authors · 2025-11-26
>
> Dear Reviewer RtrU,
>
> Thank you again for your recognition and the valuable feedback!
>
> We would greatly appreciate it if you could kindly review our responses to ensure they adequately address your concerns. We remain fully committed to clarifying any remaining points and welcome any further discussions :)
>
> Thank you for your time and consideration.
>
> Sincerely,
>
> Authors of 1067

---

> > ### Comment · Reviewer_RtrU · 2025-11-26
> >
> > Thanks for the authors' explanations, and I would keep my acceptance recommendation for this paper.

---

> > > ### Author Response · Authors · 2025-11-27
> > >
> > > Dear Reviewer RtrU,
> > >
> > > Thank you very much for your constructive feedback and your recognition of our work!
> > >
> > > Sincerely,
> > >
> > > Authors of 1067

---

### Official Review · Reviewer_vJ8R · 2025-10-27

**Soundness:** 3
**Presentation:** 3
**Contribution:** 2
**Rating:** 6
**Confidence:** 4

**Summary:**

This paper introduces VideoMind, a novel video-language agent designed to address the challenges of temporal-grounded reasoning in long-form videos. The authors identify that existing methods struggle to explicitly localize and revisit relevant video segments, a process that is natural for human cognition. The system decomposes the complex task of video reasoning into four distinct roles: a Planner to coordinate the workflow, a Grounder to localize temporal moments, a Verifier to assess and refine these moments, and an Answerer to synthesize the final response. This structured approach mimics a human-like process of breaking down problems, identifying evidence, and confirming details. To efficiently implement this multi-role system, the authors propose a novel inference-time strategy. A single base Large Multimodal Model (LMM) is equipped with multiple, role-specific Low-Rank Adapters (LoRAs). During inference, the agent can seamlessly and efficiently switch between roles (Planner, Grounder, Verifier) by activating the corresponding LoRA adapter, avoiding the significant memory overhead of loading multiple full models.

**Strengths:**

1.The core idea of decomposing video reasoning into an agentic workflow of Planner, Grounder, Verifier, and Answerer is highly intuitive and effectively addresses the limitations of monolithic, end-to-end models. It formalizes a "re-watching" and "self-verification" loop that is crucial for complex video understanding.

2. The Chain-of-LoRA mechanism is a standout contribution. It provides a highly practical solution to the problem of test-time scaling for multi-capability agents. By caching LoRA adapters in memory, it achieves the performance and flexibility of a distributed, multi-model system while maintaining the memory efficiency of a single model. The ablation study in Table 7 provides compelling evidence for this, showing it matches the performance of the "All-Distributed" approach with a fraction (≈1/4) of the memory cost.

3. The individual components are well-designed. The Grounder's timestamp decoder, which moves beyond simple text-based timestamp generation, and the Verifier's "zoom-in" strategy with special boundary tokens (<SEG-START>, <SEG-END>) are thoughtful architectural choices that contribute directly to the model's strong grounding performance.

**Weaknesses:**

1. The Planner currently relies on selecting one of three pre-defined reasoning plans based on the query. This approach may be brittle and lack flexibility when faced with novel tasks or complex queries that require a hybrid or dynamically generated sequence of steps. The paper does not explore how the system would perform on tasks that do not fit neatly into these templates.

2. The roles are trained separately on curated datasets, with the Answerer role not being fine-tuned at all. This multi-stage training process, while effective, may be complex to replicate and sensitive to the quality and composition of the data for each stage. A deeper discussion on the challenges of this approach versus a more integrated, joint-training strategy would be insightful.

3.The pipeline is inherently sequential. An error in an early stage can propagate and lead to a complete failure. For example, if the Planner mischaracterizes the query and chooses the wrong plan, or if the Grounder fails to identify any plausible moments, the subsequent steps are rendered useless. The paper could benefit from a discussion of the system's robustness and any potential fallback mechanisms to handle such cascading errors.

**Questions:**

Why the authors miss the comparison with other video reasoning works, such as LongVILA-R1-7B[1], VIdeo-R1-7B[2]

[1] Scaling RL to Long Videos
[2] Video-R1: Reinforcing Video Reasoning in MLLMs

---

> ### Author Response · Authors · 2025-11-22
> **Response to Reviewer vJ8R (1/2)**
>
> Thank you for your careful review and for the insightful feedback! We are encouraged by your recognition that the core idea of agentic workflow is intuitive and effective, the Chain-of-LoRA mechanism is highly practical, and the individual components are well-designed. Below we provide detailed responses to address your concerns.
>
> ---
>
> **Q1: How would the system perform on tasks that do not fit neatly into the pre-defined reasoning plans?**
>
> Thanks for the insight! To investigate this, we evaluate our method on `MultiHop-EgoQA` [3], a Grounded VideoQA dataset highlighting **multi-hop temporal reasoning**. For each question, the model must ground and reason on multiple relevant moments before answering, which is a paradigm that does not fit neatly into the **pre-defined single-step grounding pipeline**.
>
> | Method | Size | &nbsp;&nbsp;FT&nbsp;&nbsp; | IoU\@0.3 | mIoU | Sent. Sim. | GPT-4o Score |
> |-|:-:|:-:|:-:|:-:|:-:|:-:|
> | GPT-4o | -- | | 12.0 | 12.2 | 73.7 | **5.4** |
> | InternVL2 [4] | 8B | | 6.3 | 6.6 | 71.9 | 4.5 |
> | LLaVA-NeXT-Video [5] | 7B | | -- | -- | 62.1 | 4.2 |
> | TimeChat [6] | 7B | | 3.0 | 3.6 | 58.9 | 3.3 |
> | VTimeLLM [7] | 7B | | 8.8 | 9.2 | 70.5 | 4.3 |
> | GeLM [3] | 7B | ✓ | 18.2 | 16.7 | 75.0 | 4.8 |
> | VideoMind (Ours) | 2B | | 23.2 | 17.8 | 58.8 | 3.5 |
> | VideoMind (Ours) | 7B | | **25.1** | **19.0** | **77.3** | 4.9 |
>
> VideoMind's timestamp decoder can **predict multiple candidate moments in a single grounding step**, allowing it to effectively capture the multi-hop evidence required by this benchmark. As a result, VideoMind exhibit **strong zero-shot capabilities** on this benchmark, surpassing all open-source baselines and remaining competitive to GPT-4o.
>
> The results and discussion above have been added to `Sec. B.2` and `Table 10`.
>
> ---
>
> **Q2: A deeper discussion on the challenges of separate training vs. joint training would be insightful.**
>
> Thank you for the suggestion! We acknowledge that training roles separately on curated datasets inevitably introduces additional complexity. Nevertheless, this strategy is justifiable as the key capabilities of different roles (*i.e.,* planning, grounding, verifying, and answering) may sometimes **conflict with one another**. We have provided the performance of joint training under the `All-in-One` setting in `Table 7`, which we present below for your reference.
>
> | &nbsp;&nbsp;&nbsp;&nbsp;&nbsp;Method&nbsp;&nbsp;&nbsp;&nbsp;&nbsp; | NExT-GQA$_{mIoU}$ | NExT-GQA$_{Acc}$ | Charades-STA$_{R\@0.5}$ | Charades-STA$_{mIoU}$ | Video-MME$_{All}$ | Video-MME$_{Long}$ |
> |:-:|:-:|:-:|:-:|:-:|:-:|:-:|
> | All-in-One | 28.0 | 70.5 | 47.8 | 42.1 | 53.6 | 43.6 |
> | Chain-of-LoRA | **28.6** | **71.4** | **51.1** | **45.2** | **55.4** | **46.3** |
>
> The results clearly show that **Chain-of-LoRA offers the best balance between effectiveness and efficiency**. It consistently outperforms the `All-in-One` joint training strategy across all benchmarks, with minimal modifications to the base model. We believe that this is the most efficient way to unlock complementary capabilities without introducing much complexity.
>
> The discussion above has been added to `Sec. 4.2`.

---

> ### Author Response · Authors · 2025-11-22
> **Response to Reviewer vJ8R (2/2)**
>
> **Q3: The paper could benefit from a discussion of the system's robustness and any potential fallback mechanisms to handle cascading errors.**
>
> Thanks for the valuable insight! We conduct a systematic quantitative analysis of error propagation on two representative datasets: `ReXTime` (more temporal-related) and `NExT-GQA` (more reasoning-related). For both datasets, each error case is categorized into one of the following types:
>
> 1. **Planning Error:** The question can only be correctly answered by switching to another reasoning plan (*e.g.,* from "all roles" to "answerer only").
> 2. **Grounding Error:** Not in `1`, but all the top-5 predicted moments are incorrect (*i.e.,* temporal IoU < 0.5).
> 3. **Verification Error:** Not in `1` + `2`, but the moment selected after verification is incorrect.
> 4. **Answering Error:** Not in `1` + `2` + `3`, but the predicted answer is incorrect.
>
> Case distribution on `ReXTime` dataset:
>
> | Method | Size | Correct | Planning Error | Grounding Error | Verification Error | Answering Error |
> |:-:|:-:|:-:|:-:|:-:|:-:|:-:|
> | VideoMind | 2B | 69.1% | 1.2% | 18.3% | 5.7% | 5.7% |
> | VideoMind | 7B | 74.6% | 1.1% | 15.0% | 4.6% | 4.7% |
>
> Case distribution on `NExT-GQA` dataset:
>
> | Method | Size | Correct | Planning Error | Grounding Error | Verification Error | Answering Error |
> |:-:|:-:|:-:|:-:|:-:|:-:|:-:|
> | VideoMind | 2B | 71.2% | 1.9% | 14.0% | 6.9% | 6.0% |
> | VideoMind | 7B | 76.6% | 0.7% | 11.8% | 5.8% | 5.1% |
>
> Several conclusions can be drawn from the distributions:
>
> 1. **The planner is highly reliable**, with less than 2% error rate on both datasets.
> 2. **Grounding errors account for roughly half of all failures.** This is aligned with our hypothesis that accurate temporal grounding plays a crucial role in our proposed pipeline.
> 3. **Verification and answering contribute comparably smaller portions of the failures**, accounting for only about 5% error rate each.
>
> Overall, these results highlight that the temporal grounding stage (grounder + verifier) dominates the pipeline's error accumulation. Strengthening grounding capabilities and introducing appropriate reflection mechanisms are potentially the most effective paths toward improving end-to-end long video reasoning.
>
> The experiments and discussion above have been added to `Sec. B.3`, `Table 19`, and `Figure 7` **(Error Distribution Visualization)**.
>
> ---
>
> **Q4: Missing comparison with other video reasoning models, e.g., LongVILA-R1-7B [1] and Video-R1-7B [2].**
>
> Thanks for highlighting these important baselines! We acknowledge that these are strong video reasoning models with **pure text-based reasoning chains**, which are fundamentally different from our **vision-centric** reasoning mechanism. Below, we compare them with our VideoMind-7B on the challenging `CG-Bench`, `MLVU`, `LVBench`, `Charades-STA`, and `ActivityNet-Captions` datasets.
>
> | Method | &nbsp;&nbsp;&nbsp;Base Model&nbsp;&nbsp;&nbsp; | CG-Bench$_{Acc}$ | MLVU$_{Acc}$ | LVBench$_{Acc}$ | Charades-STA$_{mIoU}$ | ActivityNet-Captions$_{mIoU}$ |
> |-|:-:|:-:|:-:|:-:|:-:|:-:|
> | LongVILA-R1 [1] | LongVILA-7B | 26.7 | 56.5 | 34.7 | 30.0 | 21.4 |
> | Video-R1 [2] | Qwen2.5-VL-7B | 34.4 | 63.1 | 38.4 | 34.9 | 28.0 |
> | VideoMind (Ours) | Qwen2-VL-7B | **38.4** | **64.4** | **40.8** | **50.2** | **33.3** |
>
> Our method significantly outperforms both baselines on all benchmarks. These comparisons suggest that **vision-centric reasoning is superior to pure text-based reasoning on long/complex videos**.
>
> The comparison and discussion above have been added to `Sec. 2` (Related Work) and `Table 17`.
>
> ---
>
> **References:**
>
> [1] Scaling RL to Long Videos, NeurIPS 2025
>
> [2] Video-R1: Reinforcing Video Reasoning in MLLMs, NeurIPS 2025
>
> [3] Grounded Multi-Hop VideoQA in Long-Form Egocentric Videos, AAAI 2025
>
> [4] https://internvl.github.io/blog/2024-07-02-InternVL-2.0/, 2024
>
> [5] https://llava-vl.github.io/blog/2024-04-30-llava-next-video/, 2024
>
> [6] TimeChat: A Time-sensitive Multimodal Large Language Model for Long Video Understanding, CVPR 2024
>
> [7] VTimeLLM: Empower LLM to Grasp Video Moments, CVPR 2024

---

> ### Author Response · Authors · 2025-11-26
>
> Dear Reviewer vJ8R,
>
> Many thanks again for your thoughtful feedback!
>
> We would greatly appreciate it if you could kindly review our responses to ensure they adequately address your concerns. We remain fully committed to clarifying any remaining points and welcome any further discussions :)
>
> Thank you for your time and consideration.
>
> Sincerely,
>
> Authors of 1067

---

### Author Response · Authors · 2025-12-02
**Final Remark from the Authors**

Dear AC and Reviewers,

We sincerely thank you for your thoughtful review and insightful feedback, which have significantly strengthened our work.

We are encouraged by the **all-positive ratings (8666)** and the reviewers' recognition that:

- **`vJ8R`** The core idea of decomposing video reasoning into an agentic workflow is *highly intuitive*, effectively addressing the limitations of monolithic models. The Chain-of-LoRA mechanism is a *standout contribution*. The individual components are *well-designed*.
- **`RtrU`** Chain-of-LoRA is a *highly practical solution* to a major challenge in agentic AI. It directly addresses the need for multi-skilled agents *without the prohibitive memory overhead*. The claims are backed by comprehensive experiments across 14 benchmarks. The results are *consistently strong*.
- **`3Ldi`** The method is *structurally clear* and *engineering-efficient*. One model with multiple LoRAs is *an appealing design*. The grounder's technique is *solid*. The verifier is with *clear training and inference definitions*. The Chain-of-LoRA mechanism realizes *efficiency–effectiveness trade-off*.
- **`ocUG`** The Chain-of-LoRA mechanism is a *key strength*, achieving a strong balance between performance and memory efficiency. The paper is supported by comprehensive evaluation spanning different tasks. The claims are *well-substantiated* by detailed ablation studies.

Following the constructive comments and insightful suggestions, we have made the following improvements:

- **Comparison with More Baselines (`L121`, `Table 5`, `Table 17`):** As suggested by reviewers vJ8R and ocUG, we compare our method with more baselines for video reasoning (LongVILA-R1 and Video-R1) and video temporal grounding (TRACE, Grounded-VideoLLM, LLaVA-ST, and UniTime) to demonstrate its effectiveness.
- **New Evaluation Results on MultiHop-EgoQA Dataset (`L1012`, `Table 10`):** Inspired by reviewer vJ8R, we evaluate our method on this new dataset to highlight its multi-hop temporal reasoning capabilities. VideoMind exhibits strong zero-shot performance, surpassing all open-source models and remaining competitive to GPT-4o.
- **Generalizability on Domain-Specific Datasets (`L1025`, `Table 12`):** In response to reviewer 3Ldi, we provide a variant of VideoMind co-trained on additional 67K samples from NaQ dataset. This variant outperforms the domain-specific baseline trained on 1.8M egocentric samples, highlighting its strong generalizability.
- **Reliability Analysis of Each Role (`L1131`, `L1222`, `L1235`, `Table 18`, `Table 22`, `Table 23`):** Thanks to the suggestions from reviewers ocUG and 3Ldi, we provide in-depth studies of the reliabilities of the planner, grounder, and verifier to demonstrate their significance.
- **Inference-Time Efficiency Comparison (`L1242`, `Table 24`):** As suggested by reviewers RtrU and 3Ldi, we compare the average inference time per video on CG-Bench with LongVILA-R1 on a single NVIDIA RTX 6000 Ada GPU. With the auto-planning capability activated, our method is 7.8% faster than the baseline.
- **Quantitative Analysis of Error Propagation (`L1249`, `Table 19`, `Figure 7`):** In response to reviewers vJ8R and 3Ldi, we quantify the error patterns and conduct a systematic analysis of error propagation on ReXTime and NExT-GQA datasets. The results show that grounding errors are the primary source of failures.
- **Correlation between Grounding IoU and QA Accuracy (`L1264`, `Figure 8`):** As suggested by reviewer 3Ldi, we visualize the averaged QA accuracy under different grounding IoU ranges on ReXTime and NExT-GQA datasets. A clear positive correlation can be observed on the more temporal-related ReXTime dataset.
- **Controlled Experiments on Closed-Source Models (`L1270`, `Table 25`):** To address reviewer 3Ldi's concern, we demonstrate our method's effectiveness against GPT-4o and Gemini-1.5-Pro on subsets of MLVU and LVBench under strictly controlled settings (frame resolution & sampling method, max frame count, and generation configs).
- **Compatibility with Other Base Models (`L1282`, `Table 26`):** Thanks to the suggestions from reviewer RtrU, we implement the Chain-of-LoRA pipeline on two additional open-sources LMMs (InternVL3 and Qwen2.5-VL) and observe that the overall performance could be further enhanced with these stronger base models.
- **Adaptation of the Multi-Role Pipeline on Closed-Source Models (`L1287`, `Table 27`):** Following reviewer RtrU's insight, we simulate our reasoning pipeline via a series of prompts for GPT-4o, GPT-5, and Gemini-2.5-Pro. Results on subsets of MLVU and LVBench validate its effectiveness even without role-specifc model designs or training.

Other details such as providing more discussions and uploading data generation scripts are also included in the revision. We deeply appreciate all your efforts in reviewing our paper and hope that our work will make a meaningful contribution to the community.

Sincerely,

Authors of Paper 1067

---

### Meta-Review · Area_Chair_pYfF · 2026-01-10

**Summary:**

The four reviewers agree that this is a well-executed and practical framework for temporal video reasoning. The system uses a Chain-of-LoRA design to coordinate four specialized roles: a Planner, Grounder, Verifier, and Answerer. **Reviewer vJ8R**, **Reviewer RtrU**, **Reviewer 3Ldi**, and **Reviewer ocUG** all praise the engineering quality and the memory efficiency shown across 14 benchmarks. This approach allows the system to scale specialized capabilities without the high memory costs of full fine-tuning.

Initial concerns focused on the flexibility of the planning templates and the risk of errors cascading through the sequential pipeline. **Reviewer RtrU** and **Reviewer ocUG** also questioned the wall-clock latency and the dependence on Qwen2-VL. During the rebuttal, the authors added comparisons to LongVILA-R1 and Video-R1 along with a detailed error propagation analysis. They also tested the pipeline on other backbones like InternVL3 and Qwen2.5-VL to show generality. The results show 2-5% gains even when layering the pipeline on top of GPT-4o, GPT-5, and Gemini-2.5-Pro. These additions make the method look robust and broadly applicable.

**Reviewer Concerns:**

**Reviewer vJ8R** raised concerns about the rigid Planner and the complexity of the training procedure. The rebuttal keeps the original Planner design but shows that auto-planning improves runtime. The authors also provided an error analysis. It shows that 15% of failures stem from the Grounder. This suggests that errors do not cascade in an uncontrolled way. The system still relies on a limited plan space, but the empirical performance is hard to ignore.

**Reviewer RtrU** focused on efficiency and backbone dependence. The new timing results show the pipeline is 8.9% slower in the basic setting. However, the auto-planning mode is actually 7.8% faster. The authors also tested the system on InternVL3 and Qwen2.5-VL. Gains of 2-5% on GPT-5 and Gemini-2.5-Pro show the architecture adds value beyond simple prompting. These results address the main technical questions.

**Reviewer 3Ldi** asked for comparisons to newer models and more detail on error propagation. The authors added results for LongVILA-R1 and Video-R1. VideoMind outperforms both baselines. The breakdown of error sources clarifies how the pipeline fails. This is a strong engineering realization of a modular agent even if the conceptual framework is not entirely new. The performance gap on domain-specific datasets is also reduced.

**Reviewer ocUG** questioned the Timestamp Decoder and the reliance on GPT-4o for query rephrasing. The rebuttal provides better insight into module reliability. It shows that Chain-of-LoRA generalizes across backbones and improves strong closed models. This reduces the worry that the system depends too much on external tools. The questions about the Timestamp Decoder complexity are not fully settled, but they are now secondary to the performance gains.

**Reviewer Scores:**

**Reviewer vJ8R** (Original: 6 → Predicted: 6): The rebuttal adds useful analysis on error propagation and backbone experiments. These results ease concerns about brittleness. The core Planner design remains the same, so the score will likely stay at a weak accept.

**Reviewer RtrU** (Original: 8 → Predicted: 8): The authors addressed wall-clock time and backbone dependence directly. The new results are favorable and show the system is efficient. I expect the reviewer to maintain their positive stance.

**Reviewer 3Ldi** (Original: 6 → Predicted: 6): The new comparisons to Video-R1 and closed models address the empirical concerns. The reservations about conceptual novelty may prevent a score increase. A weak accept is the most likely outcome.

**Reviewer ocUG** (Original: 6 → Predicted: 6): The rebuttal offers enough evidence of generalization to support the current score. Some minor questions about query rewriting remain. The reviewer is likely to keep their weak accept.

---

### Decision · Program_Chairs · 2026-01-26

Accept (Poster)